# Quantifying nutrient fluxes in Hyporheic Zones with a new Passive Flux Meter (HPFM)

Julia Vanessa Kunz[1*], Michael D. Annable[2], Jaehyun Cho[2], Wolf von Tümpling[1], Kirk Hatfield[2], Suresh Rao[3],

Dietrich Borchardt[1], Michael Rode[1]

[1]Helmholtz Centre for Environmental Research UFZ, Magdeburg, Germany

[2]University of Florida, Gainesville, Florida (USA)

[3]Purdue University, Lafayette, Indiana (USA)

*Correspondence to:* Julia Vanessa Kunz (vanessa.kunz@ufz.de)

**Abstract.** The hyporheic zone is a hotspot of biogeochemical turnover and nutrient removal in running waters. However, nutrient fluxes through the hyporheic zone are highly variable in time and locally heterogeneous. Resulting from the lack of adequate methodologies to obtain representative long-term measurements, our quantitative knowledge on transport and turnover in this important transition zone is still limited. In groundwater systems passive flux meters, devices which simultaneously detect horizontal water and solute flow through a screen well in the subsurface, are valuable tools for measuring fluxes of target solutes and water through those ecosystems. Their functioning is based on accumulation of target substances on a sorbent and concurrent displacement of a resident tracer which is previously loaded on the sorbent. Here we evaluate the applicability of this methodology for investigating water and nutrient fluxes in hyporheic zones. Based on laboratory experiments we developed hyporheic passive flux meters (HPFM) with a length of 50 cm which were separated in 5-7 segments allowing for vertical resolution of horizontal nutrient and water transport. The HPFMs were tested in a seven day field campaign including simultaneous measurements of oxygen and temperature profiles and manual sampling of pore water. The results highlighted the advantages of the novel method: With HPFM, cumulative values for the average N and P flux during the complete deployment time could be captured. Thereby the two major deficits of existing methods are overcome: First, flux rates are measured within one device instead of being calculated from separate measurements of water flow and pore-water concentrations. Second, time integrated measurements are insensitive to short term fluctuations and therefore deliver more representative values for overall hyporheic nutrient fluxes at the sampling site than snapshots from grab sampling. A remaining limitation to the HPFM is the potential susceptibility to biofilm growth on the resin, an issue which was not considered in previous passive flux meter applications. Potential techniques to inhibit biofouling are discussed based on the results of the presented work. Finally, we demonstrate how HPFM measurements can be used to explore hyporheic nutrient dynamics, specifically nitrate uptake rates, based on the measurements from our field test. Being low in costs and labor effort, many flux meters can be installed in order to capture larger areas of river

beds. This novel technique has therefore the potential to deliver quantitative data which is required to answer unsolved questions about transport and turnover of nutrients in hyporheic zones.

Keywords: hyporheic exchange, nutrient fluxes, quantitative methods, running waters, stream metabolism, tracer dilution, ion exchange resin

# 1 Introduction

Rivers export high loads of nitrogen from inland catchments to the marine environment. The ecological and economic problems caused by eutrophication of coastal and riverine ecosystems have been recognized years ago (Patsch and Radach, 1997; Artioli et al., 2008; Skogen et al., 2014). Decades of nutrient studies have unveiled, that rivers cycle rather than only transport nutrients (Garcia-Ruiz et al., 1998a; Seitzinger et al., 2002; Galloway et al., 2003). In agriculturally dominated areas, in-stream processes may for example retain up to 38 % of nitrate ($NO_3^-$) and 48% of soluble reactive phosphate (SRP) inputs (Mortensen et al. 2016). The hyporheic zone, the subsurface region of streams and rivers that exchanges water, solutes and particles with the surface (Valett et al., 1993) and may mix stream-water during the transport through the sediments with underlying groundwater (Triska et al., 1989; Fleckenstein et al., 2010; Trauth et al., 2014), is one key compartment for instream nutrient cycling (Fischer et al., 2005; Zarnetske et al., 2011b; Basu et al., 2011; Stewart et al., 2011). For instance, denitrification, the anaerobic reduction of $NO_3^-$ to gaseous $N_2$ and in most river systems the dominant dissimilatory process which removes N out of the system (Laursen and Seitzinger, 2002; Bernot and Dodds, 2005; Lansdown et al., 2012; Kunz et al., 2016), often exclusively happens at "reactive sites" in the hyporheic zone (Duff and Triska, 1990; Rode et al., 2015). In addition to biological nutrient uptake, intermediate physical storage in the hyporheic zone disperses the propagation of pollutant and nutrient spikes which could be harmful for receiving water bodies (Runkel, 2007; Brookshire et al., 2009; Covino et al., 2010; Findlay et al., 2011). For those reasons, it is of interest to quantify the amount of nutrients reaching the reactive sites in the subsurface and the processes they undergo there (Seitzinger et al., 2006; Zarnetske et al., 2012). Transport rates of water and nutrients from the surface to the subsurface could be attributed to water levels, sediment properties and various other hydrological, biological, chemical and physical factors (Böhlke et al., 2009; Boano et al., 2014; Trauth et al., 2015). The complex interactions between these influencing factors and the temporal variability and local heterogeneity of hyporheic processes often cause high uncertainties in quantitative models. However, due to methodological restrictions, experimental investigations of nutrient dynamics in the hyporheic zone are rare and commonly exclusively of a qualitative nature (Mulholland et al., 1997; Grant et al., 2014).

Attempts to quantify hyporheic nutrient processing rates have primarily been based on benthic chamber and incubation experiments (Findlay et al., 2011; Kessler et al., 2012). Those laboratory (mesocosm and flume) experiments can estimate the denitrification potential of the substrates, usually via denitrification enzyme assays. However, the realized denitrification rates depend equally on environmental and hydrological conditions rather than on substrate type or denitrification potential alone (Findlay et al., 2011). Thus, owing to the high variability and

complexity of natural systems, hyporheic transport of nutrients cannot satisfactorily be mimicked in artificial set-ups (Cook et al., 2006; Grant et al., 2014).

Direct instream measurements of nutrient dynamics based on whole stream tracer injections, mass balances (McKnight et al., 2004; Böhlke et al., 2009) and more recently high resolution time series from automated sensors (Pellerin et al., 2009; Hensley et al., 2014; Rode et al., 2016a; Rode et al., 2016b)  can be used for determining general uptake rates on the reach scale, but do not allow identification of the reaction sites (hyporheic versus  in channel or algal canopies) or specific local uptake processes (Ensign and Doyle, 2006; Ruehl et al., 2007). Further, in-stream measurements do exclusively account for water which is re-infiltrating into the main stem after passage through the hyporheic zone. Under loosing conditions, where most of the nutrient-influx is flowing towards the groundwater, processing rates in the subsurface cannot be observed in the surface water. Likewise, if groundwater is contributing significantly to surface water chemistry, surface water mass balances do not characterize nutrient cycling in the hyporheic zone realistically (Trauth et al., 2014).

Conclusively, in situ assessments of hyporheic nutrient fluxes are indispensable. Hyporheic nutrient fluxes are commonly calculated from separate measurements of infiltration rates and pore-water concentrations (Kalbus et al., 2006; Ingendahl et. al, 2009). The exchange of water between the surface and subsurface is traditionally derived from hydraulic head differences or tracer injections (Fleckenstein et al., 2010; USEP, 2013). Time series of high-resolution vertical temperature profiles have efficiently been used to derive vertical Darcy velocity ($q_y$) (m d$^{-1}$) in the streambed. While measurements of vertical Darcy velocities are a valuable asset, primarily horizontal fluxes are needed to assess hyporheic transport and residence time (Binley et al., 2013; Munz et al., 2016).  Active heat-pulse tracing enables highly resolved in situ measurements of direction and velocity of hyporheic flow (Lewandowski et al., 2011; Angermann et al., 2012). These methods are profitable in shallow sediments (max.15-20 cm) and rivers with fine sediments, but may not be implementable in streams with coarser sediments.

Pore-water solute concentrations are typically determined from analysis of grab samples extracted with drive points (Saenger and Zanke, 2009; USEP, 2013). Alternatively, dialysis cells - so called peepers (Hesslein, 1976; Teasdale et al., 1995) - or gels (Krom et al., 1994), based on the diffusive equilibrium between the solute concentration in the pore-water and the receiver solution in the peeper or the gel, have been used to measure small scale solute distribution over highly resolved profiles on the mm to cm scale. These techniques provide valuable insights into the time specific conditions at the target site. However, they deliver effectively a snap shot of the highly temporally variable hyporheic zone processes, (Cooke and White, 1987) that may not be representative for the overall conditions in the system. Only repeated sampling at high frequencies and over longer timespans as conducted for example by Duff et al. (1998) can account for the short term variability. Attempting to characterize larger areas with these methods is laborious and costly.

In summary, direct quantitative methods for hyporheic flux measurements have two major deficiencies: First, separate measurements of water flow velocities and pore-water concentrations are necessary to calculate mass fluxes. This is not only labor intensive but can, due to the high temporal and spatial variability of hyporheic flow, also lead to incorrect estimates of flux rates (Kalbus et al., 2006). Second, most techniques for measuring pore-water

concentrations exclusively capture the concentration at the sampling time, which may not reflect the overall conditions at the sampling location. New, affordable and efficient methods for the long-term measurement of nutrient fluxes through the hyporheic zone are therefore required to validate and improve models (Boyer et al., 2006; Wagenschein and Rode, 2008; Alexander et al., 2009) site specific extent of nutrient processing in the hyporheic

zone (Fischer et al., 2009).

In groundwater studies, passive flux meters (PFMs) have successfully been used to quantify fluxes of contaminants (Hatfield et al., 2004; Annable et al., 2005; Verreydt et al., 2013) through screened groundwater monitoring wells, with integral time spans ranging from days to weeks. A standard PFM consists of a cylindrical, screened PVC casing

filled with activated carbon (AC) as a porous sorbent. As long as the PFM is residing in the monitoring well, dissolved contaminants in the groundwater flowing passively through the PFM, are retained on the AC. Furthermore, the AC is preloaded with water soluble resident tracers. The horizontal water flux through the screened media can then be determined from the displacement of the resident tracers, while simultaneously the contaminant flux is quantified based on the solute mass captured on the AC. Bench-scale column and well screen experiments

with $PO_4^-$ have been conducted in the laboratory (Cho et al., 2007). Since AC did not prove effective in capturing nutrients, an anion absorbing granular resin, originally manufactured for water purification processes, was used as a sorbent for $PO_4^-$. In theory, the PFM principle should be extendable to other nutrients or other environments. For example, development of Sediment Bed passive flux meters (SBPFM) for the measurement of vertical water flow and contaminant flux in sediments has been investigated (Layton, 2015). Like the standard PFM, SBPFM employed

AC as a sorbent incorporated into a PVC casing. The main design modification between the standard PFM and the SBPFM was the use of short screen intervals to allow vertical flow through the drive-point device thus measuring both water and contaminant flux vertically through the sediments. However, specific tests for $NO_3^-$ as well as assessments under the complexities associated with hyporheic zone processes have not been conducted.
In this study we evaluate the applicability of a PFM for the measurement of horizontal nutrient fluxes in hyporheic

zones, focusing on $NO_3^-$. We hypothesized that, while the principal concept of the PFM can be maintained, several adaptations will be necessary: Most importantly, a suitable sorbent for $NO_3^-$ as target nutrient is required. The market of anion absorbing resins is large and offers a wide range of products with varying characteristics (Annable et al., 2005; Clark et al., 2005). Various criteria, like possible interference of resin compounds with the resident tracer analysis or pre-existing background nutrient loads on the resin have to be considered. Since experience on

resin behavior under field conditions are rare, we also expected unforeseen associated challenges, including for example biofouling of resin and /or nutrients. Additionally, a new deployment and retrieval procedure was developed. In hyporheic studies, under-water installation requires a technique which minimizes contamination of the PFM by surface water and results in good contact between the PFM and the sediments. Corrections for convergence and divergence of flowlines into or around the flux meter have been established in earlier studies (Klammler et al.,

2004). However, accounting for an impermeable outer casing of a flux meter is much more complicated and requires additional factors which have to be determined experimentally for each specific application (Hatfield et al., 2004;

Klammler et al., 2004; Annable et al., 2005). We therefore developed methods deploy the PFM in a way that promotes direct contact with the surrounding sediments and minimal manipulation of the natural flow pattern. Considering these requirements, we developed a modification of the PFM for application in the hyporheic zone (Hyporheic Passive Flux Meter, HPFM). Based on the results from laboratory analysis and a first field test in a nutrient rich 3$^{rd}$ order stream (Holtemme, Germany), we demonstrate prospects and remaining limitations for hyporheic nutrient studies with HPFM.

## 2 Methods

### 2.1. Construction and materials

The Hyporheic Passive Flux Meter (HPFM) consisted of a nylon mesh which was filled with a mixture of a macroporous anion exchange resin as a nutrient absorber and alcohol tracer loaded activated carbon (AC) for the water flow quantification. In the present study HPFM were constructed 50 cm long and 5 cm in diameter. A stainless steel rod in the middle assured the stability of the device (**Figure 1**).To measure vertical profiles of horizontal fluxes of both nutrient and water in the hyporheic zone, the HPFM was divided into several segments using rubber washers. Steel tube clamps were used to attach the nylon mesh to the steel rod placed in the center of the HPFM. The nylon mesh was purchased from Hydro-Bios (Hydro-Bios Apparatebau GmbH, Kiel-Holtenau, Germany) and is available in a wide range of mesh size and thicknesses. We used a mesh size of 0.3 mm. In general, meshes should be as wide as possible because very fine mesh may act as a barrier to water flow limiting infiltration of water and solutes into the HPFM (Ward et al., 2011). However, the mesh should be smaller than the finest sediments, AC or resin granules. As final step, a rope was connected to the tube clamp on the upper end of the HPFM in order to facilitate retrieval.

### 2.2. Selection and characterization of resin

The nutrient sorbent had to meet the following criteria:

1) Have a high loading capacity for $NO_3^-$ , $PO_4^-$ and competing anions ,
2) be free of compounds which could interfere with the alcohol tracer measurements (e.g. organic substances) and
3) have a low background of $NO_3^-$ and $PO_4^-$ .

A pre-selection for anion-absorbing resins which were free of organic compounds was made based on information provided by the manufacturers (Purolite®, Lewatit®, Dowex®).

### 2.2.1. Nutrient background

Nutrient background on the resins was determined by extracting and analyzing $NO_3^-$ and $PO_4^-$ from each resin (n = 3). Therefor 30 mL of 2M KCl was added to 5 g of each pure resin and rotated for 24 hours. The solution was then analyzed on a Segmented Flow Analyser Photometer (DR 5000, Hach Lange) for $NO_3^-$ at 540 nm (detection limit

0.042 mg $NO_3^-$-N $L^{-1}$) and for SRP at 880 nm (detection limit 0.003 mg P $L^{-1}$). In order to estimate the effect of background concentrations on final results in the actual field application of HPFM, the extractable background concentrations were then converted to nutrient fluxes using a Darcy flux of $q_x = 4$ m $d^{-1}$, an estimate based on hyporheic flow velocities which were measured previously with salt tracer tests at the study site. Likewise, the expected hyporheic nutrient flux was computed from previously examined concentrations in pore water samples and the Darcy flux. The only resin with nutrient background below 5 % of expected concentrations was Purolite® A500 MB Plus (Purolite GmbH, Ratingen, Germany), which had extractable background $NO_3^-$ of 8 µg $NO_3^-$-N $g^{-1}$ wetted resin ( ± 1.6 µg $g^{-1}$, n = 3) and 0.08 µg $PO_4^-$-P $g^{-1}$ resin (± $1.7 \times 10^{-3}$ µg $g^{-1}$, n = 3). Purolite® A500 MB Plus was then considered for testing the loading capacity. The limit of quantification $LQ$ for the nutrient extraction resulting from this background was calculated according to the EPA Norm 1020B (Greenberg et al., 1992) as the sum of background concentration and 10 times the standard deviation and amounted to 24 µg $NO_3^-$-N $g^{-1}$ resin and 0.097 µg $PO_4^-$-P $g^{-1}$ resin.

### 2.2.2. Loading capacity and biofouling

Purolite® A500 MB Plus is a macroporous anion exchanger on the basis of polyvinylbenzyl-trimethylammonium with a typical granular size of 0.88 mm diameter, an average density of 685 g $L^{-1}$ and an effective porosity of 63 %. The theoretical absorbing capacity is indicated in the product sheet as 1.15 eq $L^{-1}$ (molar weight equivalences per liter of resin), corresponding to 71.3 g $NO_3^-$-N $L^{-1}$. Assuming hyporheic flow velocities of $q_x = 4$ m $d^{-1}$ and a concentration of 10 mg $NO_3^-$-N $L^{-1}$, the volume of one HPFM could adsorb $NO_3^-$ for 89 days. However, if multiple anions are present, real loading capacities for $NO_3^-$ are expectedly lower.

For the determination of a realistic loading capacity, three 5 cm diameter columns were filled to a height of 5 cm with wetted Purolite® A500 MB Plus resin, placed in a vertical position and infiltrated with water collected from the study reach. The columns were covered with tin foil to keep them dark and ensure stable temperature. A constant supernatant of 1 cm was kept on all three columns to ensure uniform infiltration at the surface of the column. Water was continuously pumped (peristaltic pump, ISMATEC® BVP Standard, ISM444) through the columns from top to bottom for 22 days at a speed of 20 mL $h^{-1}$, which also equals the expected Darcy velocity of $q_x = 4$ m $d^{-1}$. River water was supplied from a 22 L HDPE canister (Rotilabo® EPK0.1). SRP and $NO_3^-$ concentrations in this reservoir were revised daily. The draining water at the bottom outlet of the columns was sampled twice a day and analyzed for SRP and $NO_3^-$.

Biofilm growth on the resin was assessed by repeating the same experiment in smaller columns (n = 3) and extending it for several days after break-through occurred. That way, nutrient consumption by biofilm after the exhaustion of the loading capacity could be monitored. After the experiment we colored samples (n = 3) of resin granules from the columns with SybrGreen ($C_{32}H_{37}N_4S^+$) on nucleic acid and examined them under a confocal laser scanning microscope to depict the degree of bacterial fouling on the granular surface.

## 2.3. Preparation of activated carbon with alcohol tracers

As designed for the groundwater PFMs, silver impregnated activated carbon (AC) was used as sorbent for the resident alcohol tracers. The same AC as in previous PFM applications (Annable et al.,2005) was used for the HPFM in this study and was provided by the University of Florida, Gainesville. The AC had a bulk density of 550 g L$^{-1}$, a grain size ranging from 0.42 to 1.68 mm and a hydraulic conductivity $k = 300$ m d$^{-1}$.

Since the magnitude of water flow through the flux meter is unknown a priori, multiple resident tracers with a wide range of tracer elution rates were needed. The retardation factor of a substance $R_d$ is a measure for the rate of elution of the substance from a particular carrier. Alcohols offer a wide range of retardation factors and can easily be mixed and sorbed to the AC (Hatfield et al., 2004; Cho et al., 2007). By choosing the same manufacturer for the AC and the same alcohol mixture as used in the above mentioned studies, we could rely on physical and chemical characterizations and calculated $R_d$ for alcohol partitioning behavior which have been established by Hatfield et al. (2004), Annable et al. (2005) and Cho et al. (2007) (**table 1**).

An alcohol tracer mixture for approximately 10 HPFM was prepared by combining 100 mL of methanol, 100 mL of ethanol, 200 mL of isopropanol (IPA), 200 mL of tert-butanol (TBA) and 66 mL of 2, 4-dimethyl-3-pentanol (2,4 DMP).

In order to prepare the resident alcohol tracers on the AC, the AC was soaked in an aqueous solution containing the resident alcohol tracers. A standard ratio of 13 mL tracer mixture was added to 1 L of water in a Teflon sealed container and was then shaken by an automated shaker over a period of several hours. Subsequently, 1.5 L of dry activated carbon was added to the aqueous tracer solution and rotated for 12 h to homogenize the AC tracer mixture. After mixing, the supernatant water was discarded and the AC tracer mixture was stored in a sealed container and refrigerated, preventing the evaporation of the alcohol tracers

Similarly to the resins, the AC was tested for background nutrients by extraction with 30 ml KCl per 5 g AC. The activated carbon contained 0.01 mg PO$_4^-$-P g$^{-1}$AC ($\pm 7.5\times10^{-4}$ mg g$^{-1}$, n = 3) and 0.08 mg NO$_3^-$-N g$^{-1}$ AC ($\pm 5\times10^{-3}$ mg g$^{-1}$, n = 3), which amounts to 75 % of the expected concentration for NO$_3^-$ and 320 % for PO$_4^-$. To investigate whether the AC could be cleaned by washing, we repeatedly treated AC samples with distillated water or KCl as depicted in the extraction description above. Nutrients did not leach off under water treatment and neither did KCl treatment satisfactorily reduce extractable background concentration on the AC. After the third washing of AC with KCl, 0.02 mg PO$_4^-$-P ($\pm 3.3 \times10^{-4}$ mg g$^{-1}$, n = 3) and 0.04 mg NO$_3^-$-N ($\pm 2.3\times10^{-3}$ mg g$^{-1}$, n = 3) could still be extracted per g AC. Further, it was unclear to which degree replacing absorbed nutrients by KCl would alter the alcohol tracer retardation and extraction on the AC. For those reasons, we decided to keep the nutrient absorbing resin separated from the AC. As AC did not release background nutrients under water treatment, water flowing first through AC and afterwards resin layers was not considered problematic.

## 2.4. Deployment and retrieval procedure

HPFM were built, stored dry and transported in 70 cm long standard polyethylene (PET) tubes (58 x 5.3 SDR 11) purchased from a local hardware store (Handelshof Bitterfeld GmbH, Bitterfeld, Germany). To avoid resident alcohol tracer loss, the transport tubes with the HPFMs were sealed with rubber caps and cooled during storage and transport. In the field, prior to installation, the HPFMs were transferred to a stainless steel tube, 5.3 cm inner

diameter with a loose steel drive point tip on the lower end. The diameter of the steel tube for installation tightly fitted with the rubber washers at the top and bottom end of the HPFM, so that vertical water flow through tube and HPFM during installation was inhibited. The steel casing and HPFM were driven into the river bed using a 2 kg hammer until the upper end of the HPFM was at the same level as the surface-subsurface interface. The metal casing was retrieved while the HPFM was held in place using a steel rod.

After 7 days of exposure, the HPFMs were retrieved by holding the transport tube in place and quickly drawing the HPFM into the tube using the rope fixed to the upper end of the HPFM. The required length of the transport tube, steel drive casing and retrieval rope was determined by the depth of the water level in the stream.
After retrieval, the HPFM were transported to the laboratory.

## 2.5. Analysis and data treatment

In the laboratory, the retrieved HPFM were quickly (after maximal 12 hours) sampled for analysis. Therefor one segment after the other was cut open and the sorbent was segment-wise recovered, homogenized and a subsample transferred to 40 mL glass vials. The subsamples from resin segments were then analyzed for nutrient content, the subsamples from AC segments were analyzed for the remaining alcohol tracers as described in the following

paragraphs.

### 2.5.1. Water flux

The AC samples were shipped to the University of Florida for analysis. In the laboratory, the mass of the previously applied mixture of alcohol tracers in standard AC samples and the tracer mass remaining in the final AC samples were extracted with iso-butyl alcohol (IBA). About 10 g of AC samples were transferred into pre-weighed 40 mL

vials containing 20 mL IBA. Vials were rotated on a Glas-Col Rotator, set at 20 % rotation speed, for 24 h. Then, subsamples were collected in 2 mL GC vials for alcohol tracer analysis. The samples were analyzed with a GC-FID (Perkin Elmer Autosystem) (Cho et al.,2007).

The relationship between time averaged specific horizontal discharge $q_x$ (m s$^{-1}$) through the device and tracer elution is given by equation (1) (Hatfield et al., 2004)

$$q_x = \frac{1.67 \, r\theta \, (1-M_R) \, R_d}{t} \tag{1}$$

where $r$ (m) is the radius of the HPFM, $\theta$ is the volumetric water content in the HPFM (m³ m⁻³), $M_R$ (-) is the relative mass of tracer remaining in the HPFM sorbent, $t$ (s) is the sampling duration and $R_d$ (-) is the retardation factor of the resident tracer on the sorbent.

### 2.5.2. Nutrient flux $J_N$

$NO_3^-$ and $PO_4^-$ were extracted and analyzed in the laboratory at UFZ in Magdeburg, Germany, similarly to the analysis of background concentrations on the resin: subsamples of 5g resin were treated with 30 mL of 2 M KCl each and rotated for 24h for extraction. The solution was then analyzed as described above.

The time-averaged advective horizontal nutrient flux $J_N$ (mg m² d$^{-1}$) can be calculated using the following relationship (Hatfield et al., 2004):

$$J_N = \frac{q_x M_N}{2\alpha r L t} \tag{2}$$

where $M_N$ (kg) is the mass of nutrient adsorbed, $L$ (m) is the length of the vertical thickness of the segment and $\alpha$ (-) is a factor ranging from 0 to 2 that characterizes the convergence ($\alpha > 1$) or divergence ($\alpha < 1$) of flow around the HPFM. If, like in the case presented here, the hydraulic conductivity of the HPFM sorbent (resin or AC) is much higher than that of the surrounding medium and the HPFM is in direct contact with the sediments (i.e. in absence of an impermeable outer casing or well wall), $\alpha$ can be estimated after Strack and Haitjema (1981)

$$\alpha = \left( \frac{2}{1+\frac{1}{K_D}} \right) \tag{3}$$

where $K_D = k_D\, k_0^{-1}$ is the dimensionless ratio of the uniform hydraulic conductivity of the HPFM sorptive matrix $k_D$ (L T$^{-1}$) to the uniform local hydraulic conductivity of the surrounding sediment $k_0$ (L T$^{-1}$). For more details on the correction factor $\alpha$ and applications where a solid casing is required or the permeability of the surrounding sediments is higher than of the device see Klammler et al. (2004) and Hatfield et al. (2004).

## 2.6. Field testing of hyporheic passive flux meters (HPFMs)

### 2.6.1. Study site

A 30 m long stretch of the Holtemme River, a 3$^{rd}$ order stream in the Bode catchment, TERENO Harz/Central German Lowland Observatory, served as study site (51°56'30.1"N, 11°09'31.8"E). The testing reach is located in the lowest part of the river, where the water chemistry is highly impacted by urban effluent and agriculture (Kamjunke et al., 2013). Long stretches have been subjected to changes in the natural river morphology by canalization (Landesbetrieb für Hochwasserschutz und Wasserwirtschaft Sachsen-Anhalt, 2009).

The sediments at the selected site are sandy with gravel and small cobbles. Sieving of sediment samples delivered the effective grain size $d_{10}$= 0.8 mm and a coefficient of uniformity $C_u$ = 3.13. The effective porosity $n_{ef}$ is 13 %. After Fetter (2001) the intrinsic permeability was estimated to be $K_i$ = 96 m² and the hydraulic conductivity $k$ = 81 m d$^{-1}$ Clay lenses are present in the deeper sediments below 35 cm.

Mean discharge in the stream is 1.35 m³ s$^{-1}$ with highest peaks around 5-6 m³ s$^{-1}$. Discharge is continuously recorded by the local authorities at the gauge Mahndorf, 15 km upstream of the testing site. In the course of the year, $NO_3^-$ concentrations in the lower Holtemme vary between 2 and 8 mg $NO_3^-$-N L$^{-1}$(Hochwasservorhersagezentrale, 2015/2016).

The equipment was installed for a period of 7 days from 4$^{th}$ to 11$^{th}$ June 2015 as illustrated in **figure 2**.

### 2.6.2. HPFM testing

Based on the laboratory results for the nutrient backgrounds and the consequent necessity to keep resin and AC separated two approaches for constructing and deploying HPFM were tested in the field.

        A)         Resin only and AC only HPFMs

HPFMs were constructed of which 2 contained only resin (R1 and R2) and the other two contained only AC (AC3 and AC4). The HPFMs were then installed in pairs: AC only and resin only next to each other with a separation distance of 30 cm. Those 4 HPFMs were sectioned in 5 horizontal flow segments, each with a vertical length of 10 cm.

For the calculation of the nutrient flux through each segment of R1 and R2, we used the corresponding water flux through the respective segment of AC3 and AC4.

        B)         Alternating segments of AC and resin HPFMs

HPFMs L5 and L6 consisted of 7 segments starting and ending with an AC segment and adjacent segments altering between resin and AC (also see **figure 1**). Each segment had a length of 7 cm.

For the calculation of the nutrient flux through the resin segments we used the interpolated water flow measured in the two adjacent AC segments.

One additional HPFM with alternating layers was used as a control HPFM, in order to assess potential tracer loss or nutrient contamination during storage, transport and deployment/retrieval. This control was stored and transported together with the other HPFMs. After deploying the control HPFM, it was immediately retrieved, transported back to the laboratory and stored until it was sampled and analyzed along with the other HPFMs. The results from the control HPFM also include uncertainties arising from sample storage, analytical processing and the background concentration of nutrients on the resin. Measurements of the other HPFMs were corrected by subtracting the transport, storage and deployment related tracer loss and nutrient accumulation detected in the control.

### 2.6.3. Additional measurements

Vertical Darcy velocity ($q_y$)

The vertical vector of hyporheic Darcy velocities $q_y$ were measured supplementary to the horizontal fluxes assessed with the HPFM in order to estimate the general direction of flow (upwards or downwards) and to calculate the angle of hyporheic flow.

The vertical Darcy velocity ($\underline{q_y}$) (m d$^{-1}$) in the streambed was calculated using temperature profiles measured between January 2015 and October 2015. According to Keery at al. (2007) and Schmidt et al. (2014), vertical flow velocities can be computed from the temporal shift of the daily temperature signal in the subsurface water relative to the surface water. A multi-level temperature sensor (Umwelt- und Ingenieurtechnik GmbH, Dresden, Germany) was installed at the test site in January 2015. Temperature was recorded at the surface-subsurface interface and at depths of 0.10, 0.125, 0.15, 0.2, 0.3 and 0.5 m in the sediment at a 10 min interval (accuracy of 0.07 °C over a range from 5

to 45 °C and a resolution of 0.04 °C). A numerical solution of the heat flow equation was then used in conjunction with Dynamic Harmonic Regression signal processing techniques for the analysis of these temperature time series. The coded model was provided by Schmidt et al. (2014).

Oxygen profiles

We monitored the subsurface oxygen concentration as a primary indication on the redox status of the hyporheic zone in order to evaluate the potential for $NO_3^-$ reduction and $PO_4^-$ mobilization. Therefor two oxygen loggers (miniDO$_2$T, Precision measurement engineering Inc.) incorporated into steel tubes acuminated at the lower end were installed in the river bed. The tubes had filter-screens at the measuring depths of 25 and 45 cm below surface-subsurface boundary. Installation was carried out 4 weeks prior to the experiments, allowing enough time for re-

equilibration of the surrounding media. The measurement time step was 5 min.

Multi-level samplers (MLS)

Pore water nutrient concentrations were measured to substantiate the HPFM results. Multi-level samplers as described in detail by Saenger and Zanke (2009) are devices for the manual extraction of hyporheic pore water from several distinct depths. The two samplers A and B used in these experiments were manufactured by the institutional

workshop of the UFZ. Like the oxygen loggers both MLS were installed 4 weeks prior to the experiment. They consisted of an outer stainless steel tube with a length of 50 cm and a diameter of 5 cm. Ceramic filters were inserted in this outer steel mantle marking the extraction depths at 5, 15, 25 and 45 cm. The inner sides of the filters were attached to steel pipes that ran to the top of the sampler so that Teflon tubes could be attached. A protective hood was threaded on the upper end of the sampler to preclude particles and sediment entrance. Per sampler and depths 10

mL of pore water was manually extracted by connecting a syringe to the open end of the Teflon tube and slowly sucking up water at a rate of 2 mL min$^{-1}$. The 4 extraction depths were sampled successively, always starting with the shallowest depths and continuing with ascendant depths. Manual pore water samples were taken on the 4$^{th}$ and 11$^{th}$ of June 2015, both times between 1 pm and 4 pm local time.

The samples were filtered in the field through a 0.45 μm membrane filter and placed in boro-silica glass vials for

transport to the laboratory. Analysis for $NO_3^-$, SRP, sulphate ($SO_4^{2-}$) and boron (B) were conducted in the central analytical laboratory of the UFZ, Magdeburg, Germany. Analytical procedure for $NO_3^-$ and SRP was done according to the description above.

$SO_4^{2-}$ and B were used as natural tracers for groundwater and surface water respectively. $SO_4^{2-}$ was analyzed on an ion chromatograph (ICS 3000, ThermoFisher, former DIONEX), B was analyzed on an inductively coupled plasma

mass spectrometer (ICP-MS 7500c, Agilent). As $NO_3^-$ and SRP concentrations in the pore water samples taken on June 4$^{th}$ and 11$^{th}$ 2015 were unexpected and inconsistent with results from the HPFMs, the sampling was repeated on the 8$^{th}$ of October. The aim of this repeated sampling was to investigate whether diurnal variations in subsurface $NO_3^-$ and SRP concentrations could explain the discrepancies between MLS and HPFM results. We assumed that the HPFM measurements integrated temporal oscillations, while MLS samples represented the specific concentrations

around noon. In order to test this hypothesis, both MLS were sampled twice, the first time in the early morning before sunrise and again in the early afternoon (around 2 pm) during the sampling in October. Those samples were analyzed for $NO_3^-$, SRP and $SO_4^{2-}$. Due to technical issues, boron could not be measured in October.

Surface water chemistry

Surface water concentrations of SRP and $NO_3^-$ were monitored in order to compare surface and subsurface water chemistry. Therefor we installed an automated UV absorption sensor for $NO_3^-$ (ProPS WW, TriOS) at the beginning of the testing reach for the duration of the experiments. The pathway-length of the optical sensor was 10 mm, measuring at wavelengths 190-360 nm with a precision of 0.03 mg $NO_3^-$-N $L^{-1}$ and an accuracy of ± 2 %. The measurement time step was set to 15 min. SRP, $SO_4^{2-}$ and B concentrations in the surface water were assessed with grab samples taken simultaneously to the MLS measurements.

The UV sensor was supplemented with a multi-parameter probe YSI 6600 V2/4 (YSI Environmental, Yellow Springs, Ohio) recording the following parameters: pH (precision 0.01 units, accuracy ± 0.2 units), specific conductivity ( precision 0.001 mS $cm^{-1}$, accuracy ± 0.5 %), dissolved oxygen (precision 0.01 mg $L^{-1}$, accuracy ± 1%), temperature (precision 0.01 °C, accuracy ±0.15 °C) and turbidity (precision 0.1 NTU, accuracy ± 2 %).

### 2.6.4. Estimates of nitrate turnover rates based on HPFM measurements

Estimates for hyporheic removal activity $R_N$ for the specific conditions at the study site during the HPFM testing phase were calculated using the morphological and hydrological parameters summarized in **table 2.**

The absolute amount of water passing the screened area of the hyporheic zone $Q_{HZ}$ (m³ $s^{-1}$) is the product of the average horizontal vector of the Darcy velocity $q_x$ (m $s^{-1}$) measured in the HPFM and the cross sectional area of the upper 50 cm of the hyporheic zone $A_{HZ}$ (m²). The proportion of water infiltrating the hyporheic zone $\%Q_{HZ}$ (%) was then calculated from the ratio $\frac{Q_{HZ}}{Q_{SW}}$, where $Q_{SW}$ (m³ $s^{-1}$) is the average discharge at the study site during the days of measurements, derived from continuous records at the gauche Mahndorf, which were provided by the local authority Landesbetrieb für Hochwasserschutz und Wasserwirtschaft Sachsen-Anhalt.

The $NO_3^-$ removal activity of the hyporheic zone $R_N$ (%) was calculated from the difference in average surface water concentration $C_{NO3-SW}$ (mg $NO_3^-$-N $L^{-1}$) and the average concentrations measured with the HPFM $C_{NO3-HZ}$ (mg $NO_3^-$-N $L^{-1}$), were $C_{NO3-HZ}$ is the quotient $\frac{J_N}{qx}$.

# 3. Results

## 3.1. Laboratory experiments

### 3.1.1. Loading capacity and biofouling

Break-through in the sorbent column experiments occurred after 300 pore volumes (PVs) or 21 days at selected drainage for both $NO_3^-$ and SRP.

In the biofouling experiment, the $NO_3^-$ concentration in the draining water gradually decreased again after break-through. SRP in the draining water was completely depleted 6 h after the break-through. The calculated amount of retained nutrient in comparison to manufacturer value loading capacities of Purolite® A 500MB Plus indicate that the absorbing capacity of the resin in this small column experiment was exhausted after 25.5 hours (APPENDIX A).

We attributed the decrease of nutrients in the draining solution after breakthrough to biotic consumption of SRP (limiting nutrient) and $NO_3^-$.Under the laser scanning microscope growth of biofilm could be observed on obviously brown stained Purolite® beads of the columns from the biofouling experiment and to a very low degree on beads from the same column which appeared still clean (APPENDIX A). Browning of Purolite® beads was not observed on Purolite® beads from the loading experiment (bigger columns, experiment not extended after break through) but on the top 2 cm of the HPFM R2 after exposure at the study site.

## 3.2. Field testing

### 3.2.1. HPFMs and additional measurements

HPFMs

Deployment required approximately 15 min per HPFM and could be conducted by two persons. The water depth during the installation was 40 to 100 cm, depending on the specific location in the stream.
The average horizontal water flow $q_x$ and nutrient flux $J_N$ measured in the HPFM during the 7 day field testing are illustrated in **figure 3**. All flux meter except 5L showed declining $J_N$ and $q_x$ with depth. Average horizontal $q_x$ was 76 cm d$^{-1}$, ranging from 115 cm d$^{-1}$ in the shallowest layer of 5L to 20 cm d$^{-1}$ in the deepest layer of AC4. Over the 7 days duration of the experiment, accumulated horizontal flow velocities of $q_x$ = 8.4 cm 7 d$^{-1}$ ($\pm$ 0.02 cm 7 d$^{-1}$,n = 3) and nutrient fluxes of 29.4 mg $NO_3^-$-N m² 7 d$^{-1}$ ($\pm$ 0.7 mg m² d$^{-1}$, n = 3) and 36.4 mg SRP m² 7 d$^{-1}$ ($\pm$ 6.3 mg m² d$^{-1}$, n = 3) were detected in the control HPFM. Breaking these results down to dial values, yields $q_x$ = 1.2 cm d$^{-1}$ ($\pm$ 0.003 cm d$^{-1}$, n = 3) and nutrient fluxes of 4.2 mg $NO_3^-$-N m² d$^{-1}$ ($\pm$ 0.1 mg m² d$^{-1}$, n = 3) and 5.2 mg SRP m² d$^{-1}$ ($\pm$ 0.9 mg m² d$^{-1}$, n = 3). Comparing these fluxes  to the $J_N$ values measured with the other HPFM, an average 0.3 % of the uncorrected $NO_3^-$ flux and 5 % of the uncorrected SRP flux were attributed to tracer loss or nutrient accumulation resulting from transport, deployment, retrieval, analytical processing of samples and the background concentrations on the  resin.

Vertical Darcy velocity ($q_y$)

Vertical water flow $q_y$ in the stream bed was predominantly downward from January to October 2015. It was exclusively downward during the HPFM testing phase, ranging from 40 to 55 cm d$^{-1}$. With this, vertical flow $q_y$ was slightly lower than average horizontal flow $q_x$. Resulting from the relationship between $q_y$ and $q_x$ the angle of hyporheic flow ($\tan\alpha = \frac{q_y}{q_x}$) was 32° downwards.

Oxygen profiles

We observed strong diel variations in oxygen concentration in the hyporheic zone. During several nights oxygen was nearly depleted (**figure 4**).The minima and maxima oxygen concentrations in the subsurface occurred contemporarily with the respective extremes in the surface water. Interestingly the amplitude in $O_2$ oscillation was higher at 45 cm depths than at 25 cm depths.

Multi-level samplers (MLS)

In order to facilitate direct comparison, nutrient fluxes as measured in the HPFM were converted to flux averaged concentrations which are the quotient of $J_N$ and the respective $q_x$ (**figure 5**). Overall, nutrient concentrations in the

manually sampled pore water taken in June 2015 were higher than the average concentration derived from the HPFM. While the expected increase of SRP and decrease of $NO_3^-$ and water flow with depths was observed in the HPFM, pore water extracted with the MLS showed no change over depth neither for $NO_3^-$ nor SRP. In the repeated manual pore water samples taken in October (**figure 6**) $NO_3^-$ concentrations were uniformly lower in the early

morning than in the afternoon, whereas SRP behaved the other way round. This trend was consistent in both samplers even though the average concentration and distribution over depths differed between the samplers A and B. On both sampling dates in June (04.06. and 11.06.2015) neither $SO_4^{2-}$ nor boron showed a vertical gradient in concentrations in the pore water samples. $SO_4^{2-}$ concentrations of 170 mg $L^{-1}$ on the 4th June and 190 mg $L^{-1}$ on the 11th June were in the same range as surface water concentrations. Likewise were boron concentrations with 50 to 60

µg $L^{-1}$ consistence with the concentrations in the surface water, indicating only minor groundwater influence. Also in October $SO_4^{2-}$ concentrations in the pore water samples were in the range of surface water concentrations, slightly declining with depth.

Surface water chemistry

Temperature, $O_2$ and pH showed the expected diurnal amplitudes whereas specific conductivity and $NO_3^-$ did not

display a distinct diurnal pattern (**table 4**).

### 3.2.2. Estimates of nitrate turnover rates based on HPFM measurements

With an average water flow of $Q_{HZ} = 2.65 \times 10^{-5}$ m³ s$^{-1}$ through the assessed upper 50 cm of the hyporheic zone and across the 6 m width of the stream, 0.008 % of water transported in the river entered the hyporheic zone (**table 3**).

While the average surface water concentration was 2.86 mg $NO_3^-$-N $L^{-1}$, the average concentration in the subsurface measured with the HPFM was only 1.39 mg $NO_3^-$-N $L^{-1}$. Assuming that the difference between surface and subsurface concentration arose from hyporheic consumption of infiltrating $NO_3^-$, the average removal rate $R_N$ was 52 %. For SRP the average surface water concentration from 4th to 11th June 2015 was 0.165 mg P $L^{-1}$, the average concentration in the hyporheic zone was 0.11 mg P $L^{-1}$.

# 4. Discussion

The application of the HPFM for quantitative in situ measurement of horizontal $NO_3^-$ and SRP fluxes through the hyporheic zone is novel. An earlier study on passive flux meter (SBPFM) in river beds (Layton, 2015) only assessed vertical flow of contaminants and is therefore not comparable to the application presented here. In the current work,

adaptations were developed, tested and improved. Those include the choice of an appropriate resin, assessment of biofilm growth on the instruments and an approach that avoids challenges with contamination of the sorbent with nutrients. The results from the control HPFM showed that the uncertainty in measurement related to handling of the HPFM and processing of samples as conducted in this study is acceptable. Finally, the minimum and maximum deployment time will depend on the Darcy velocity and nutrient concentrations at a study site. Since the values

derived from the control incorporate all the processing steps of HPFM and samples, they can be regarded as the

method detection limit *MDL* (Greenberg et al., 1992). The *MDL* defines the lower limit for the use of HPFM in cases where nutrient fluxes are very low and deployment time cannot be extended. Based on the accumulated values detected in the control, a minimum deployment time can be estimated. In systems with high nutrient concentrations, usually the flow velocity $q_x$ will be the limiting factor. In our application the *MDL* for $q_x$ derived from the control was 8.4 cm for the complete deployment time (7 days). If the method inherent uncertainty should not be more than 5 % of the total measurement, the product of duration (in days) and velocity (in cm) should be at least 168 (20 times 8.4). As an example: if measured $q_x$ is around 200 cm d$^{-1}$, one day (24 h) of deployment is sufficient. The lowest $q_x$ detected in our assessment was 21 cm d$^{-1}$ (in HPFM AC4, see figure 3f), so that a deployment duration of 8 days would have been optimal. The same estimation can also be derived for expected nutrient fluxes. In systems with low nutrient concentrations, it would be preferable to estimate the minimum deployment time based on nutrient fluxes. We recommend that a control HPFM is incorporated in each field application of HPFM in order to determine the specific *MDL*. The upper limit is given by the loading capacity of the resin or complete displacement of all resident alcohol tracers.

The high nutrient background on the AC required the separation of resin and AC in the HPFMs. We tested two different HPFM designs in this study, of which each inherits designated characteristics being more or less beneficial for different specifications: The first approach, pairs of two HPFM where one is used to assess the water flux and the second to capture nutrients is preferable if a highly resolved depth profile is needed (a heterogeneous horizontal flux in the vertical direction). Since this approach assumes that local horizontal heterogeneity is negligible in the range of 20-30 cm, we recommend this type only for the use in uniform systems such as channelized river reaches. Even in those systems however, small scale variability in stream bed and sediment characteristics can cause spatially heterogeneous flow distributions (Lewandowski et al., 2011; Mendoza-Lera and Mutz, 2013). The second approach with alternating nutrient sorbents and water flux measuring segments is therefore preferable in most other cases as long as a high resolution over the vertical profile is not required. In general, several HPFM should be grouped together in order to obtain representative results.

Further improvements of the HPFM for nutrient studies in the subsurface of rivers could be achieved by identifying a nutrient free carrier for the tracers. First, because this would allow measuring nutrient and water flux at the same location within the device and thereby increase spatial resolution. Second, because in a mixed texture of nutrient absorber and tracer carrier the antibacterial nature of the activated carbon would suppress biofouling on the absorbent. We observed substantial biofilm growth on the resin in the laboratory and on the top 2 cm of the field-deployed HPFM R2. The results of the column experiments suggest that biofilm growth on the resin porous media did not affect its loading capacity. Further, biofilm growth was only visible on columns which were run beyond breakthrough, suggesting that considerable biofouling only started after the loading capacity of the tracer was exhausted. R2 detected higher $NO_3^-$ fluxes in the top layer than the other HPFM. This could be due to contamination of the top layer of this HPFM with surface water (if the HPFM was not introduced sufficiently deep into the sediments). The further implication would be, that this layer was exposed to much higher water and nutrient infiltration, so that the loading capacity was exhausted before the end of the experiment allowing biofilm accumulation. At the current state it is unclear, to what extent the biofilm bound nutrients can be extracted by the

procedure used here. Further experiments would also be needed to clarify under which conditions biofilm growth can occur and if bacterial uptake, transformation and release of nutrients influence the concentrations of nutrients inside the HPFM. HPFM segments on which biofilm is visible should be interpreted with caution. Finally, identifying a procedure or materials which completely inhibit biofouling will be an important step in the further

development of HPFM.

In addition to instrumental adaptations we presented an installation procedure, which allows for smooth deployment with minimal disturbance of the system. Unlike typical well screen deployments where PFM (Annable et al., 2005; Verreydt et al., 2013) or SBPFM (Layton, 2015) have been inserted into a screened plastic or steel casing, our technique enabled the direct contact of the HPFM with the surrounding river sediments. Disturbing the natural

structure of the sediment, potentially resulting in artificial flow paths is intrinsic to all intrusive techniques, including HPFM. Still, dispensing of a well screen improves the installation of the HPFM in the natural system and minimizes the generation of preferential flow paths along the wall of the device. Additionally, the HPFM includes a deployment time that is long relative to the installation period, suggesting that the method generates lower disturbance compared to other intrusive measurements. While the installation of mini-drive points or heat pulse

sensors in sediments coarser than sand may be difficult or even impossible and also proved unfeasible at our field site, installation of the HPFM with the presented technique was successful. The correction for convergence of flowlines into the device or divergence around it is relatively simple and already incorporated in the equation for the flux calculation. Heterogeneous permeability of the hyporheic zone around the HPFM does not distort the correction term as long as the permeability of the surrounding media is substantially lower than the permeability of the HPFM

matrix. Pre-measurements are therefore necessary for the selection of a suitable resin and tracer carrier. We believe that the presented approach and equations are applicable for a wide range of field conditions. However, for very coarse sediments, a protection of the HPFM with a solid screen might still be preferred. If fine particles are observed to bypass the mesh and enter the HPFM, a finer mesh should be chosen. We did not observe clogging of the mesh or intrusion of particles at our study, though in highly permeable systems with fine particle transport this might have to

be considered.

A mayor advantage of the HPFM method is highlighted by the findings of the 7 day long field testing: In June, we found discrepancies between the average concentrations measured in the HPFM and the concentration found using the MLS. From our measurements it is not possible to prove that the HPFM results are correct and the MLS results biased. Nevertheless, the HPFM showed the expected decline in $J_N$ with depths, whereas the MLS pore water

concentrations were similar at all depths. This can be related to two reasons: First, we might have sampled surface water which bypassed along the wall of the MLS. The question would then be why that happened in June but not in October. Second, we might have sampled the MLS at a time point when the hyporheic zone was inactive in respect to nutrient processing. Considering the high diurnal amplitudes in hyporheic oxygen concentration, we assumed that the discrepancy between HPFM and MLS arose from oscillations in hyporheic nutrient concentrations similar to the

oxygen pattern. Microbial consumption of $O_2$ in the sediments can, depending on nutrient concentration in surface water and transfer of these nutrients to the sediments, result in $O_2$ depletion in the subsurface. Especially in nutrient rich streams the related diurnal oscillations in $O_2$ concentration favor night time denitrification in the hyporheic zone

(; Christensen et al., 1990; Laursen and Seitzinger, 2004; Harrison et al., 2005; O'Connor and Hondzo, 2008; Nimick et al., 2011). The redox conditions in the subsurface may also regulate the mobilization/demobilization of phosphate (Smith et al., 2011). The repeated manual sampling of pore water from MLS in October showed diurnal variations of SRP and $NO_3^-$ in the subsurface of the testing reach, supporting the hypothesis that diurnal cycles in

benthic metabolism caused temporal variations in hyporheic SRP and $NO_3^-$ concentrations at our study site. As the majority of sampling is commonly conducted during daylight hours, night time conditions are underrepresented in studies relying on single manual sampling events. Flux average concentrations can deviate by more than 50 % from estimates based on single event sampling, as was illustrated by comparison between our manual samples and the average pore water concentrations calculated from the HPFM data.

Repeated pore water sampling at high frequencies can be used to determine diurnal dynamics. However, continuing this over a longer time span is laborious, whereas if only few single time specific snap shot samplings are conducted, the results may not realistically represent the overall conditions at the target site. Our comparison between MLS and HPFM reinforce the need for long term recording of nutrient transport through the hyporheic zone. In general, most of our knowledge on hyporheic nutrient dynamics is based on measured surface water dynamics and models which

project these dynamics on hyporheic processing. Theoretically, we could measure nutrient fluxes in the hyporheic zone and estimate whole stream uptake rates from these measurements. However, the substantially higher effort to obtain subsurface data is not justified in most cases. As long as the overall in-stream retention is the focus, surface water monitoring will remain the method of choice. Innovative tracer experiments may even allow quantifying hyporheic exchange in streams. Haggerty et al. (2009) proposed a "smart" tracer approach, where the injected

substance resazurin converts irreversibly to resofurin under metabolic activity. While a promising tool for detecting metabolic activity at the sediment-water interface in streams, first, uncertainties about sorption and transformation characteristics of these tracers remain (Lemke et al., 2013) and second, those methods give no evidence about nutrient transport to those reactive sites.

Thus, whenever the nutrient processing function of the hyporheic zone and its quantitative contribution to stream

nutrient retention is of interest, for example in the evaluation of restauration measures including a rehabilitation of the river bed, direct measurements of hyporheic fluxes are indispensable. The HPFMs are a valuable approach that can be efficiently used to characterize and quantify nutrient dynamics in a sediment system. We consider that a combination of HPFM, MLS and concurrent measurements of pore water oxygen concentrations, as presented in this study, provide a practical set-up to interpret hyporheic nutrient dynamics.

Like solute concentrations and water flow patterns, the vertical extension of the hyporheic zone varies in time and space and between different rivers and reaches. Our set-up assessed exclusively the upper 50 cm of the hyporheic zone. We found continuously decreasing $NO_3^-$ concentrations with depths, suggesting that this entire area (and potentially deeper) of the subsurface contained active sites for nitrate removal. While it was stated that denitrification is limited to the upper few cm of the hyporheic zone close to the sediment-water interface (Hill et al.,

1998; Harvey et al., 2013), our results are in accordance to findings by Zarnetske et al. (2011b) and Kessler et al. (2012) who also report extended active hyporheic zones. Conducting collateral tracer tests, as suggested for example by Abbott et al. (2016), could deliver further evidence and characterize distinct flow paths. Nevertheless, since

vertical water movement was overall downward and the lowest concentrations of $NO_3^-$ were observed in the deepest segments of the HPFM, it is very likely that the hyporheic zone at our study site extends deeper than the 50 cm evaluated.  The length of an HPFM can easily be increased, depending on the individual site conditions. Considering the high spatial heterogeneity of the hyporheic zone, a larger number of HPFM would be needed to derive reliable and statistically supportable rates of hyporheic nutrient dynamics. The following example aims to display further possibilities of interpreting HPFM measurements. At our study site, the hyporheic removal potential $R_N$ of more than 50 % of infiltrating $NO_3^-$ and 30 % of SRP suggests an active hyporheus. Evaluation of the effect of hyporheic removal activity on overall $NO_3^-$ removal in the stream or the normalization of hyporheic uptake to a benthic area requires a flow path length. In the presented example, this length refers to the horizontal vector of the distance the water travels in the subsurface before infiltrating the HPFM. The horizontal vector can be derived from the residence time of water and solutes in the hyporheic zone $\tau_{HZ}$ and the horizontal Darcy velocity $q_x$. Assuming a downward flow direction, $\tau_{HZ}$ could be inferred from the vertical Darcy velocity $q_y$ as assessed from the temperature profiling and the hyporheic zone depths of 50 cm. Thereafter, $\tau_{HZ}$ conceptually corresponds to the time the water travels through the hyporheic zone before exiting to groundwater and $s_{HZ}$ to the horizontal vector of the flow paths. The nitrate uptake rate $U_{NO3\text{-}HZ}$ (mg $NO_3^-$-N m$^{-2}$ d$^{-1}$) is then the difference between the theoretically transported $NO_3^-$ mass $M_{NO3\text{-}HZ\ theor}$, which is the product of $Q_{HZ}$ and $C_{NO3\text{-}SW}$ and the measured mass flux $M_{NO3\text{-}HZ\ real}$. During the testing phase $U_{NO3\text{-}HZ}$ was calculated as 693 mg $NO_3^-$-N m$^{-2}$ d$^{-1}$. The same procedure yields a removal (uptake or adsorption) rate for SRP of $U_{PO4\text{-}HZ} = 24$ mg $PO_4^-$ m$^{-2}$ d$^{-1}$. Calculating $U_{NO3\text{-}HZ}$ in the same way for each single depth assessed with the HPFM can deliver additional information about vertical gradients on nutrient processing rates and help to identify the most active depth in hyporheic zone. $U_{NO3\text{-}HZi}$ of a particular layer in the hyporheic zone can be derived by the differences in uptake rate between the regarded layer and the overlying layer. For instance the removal rates attributed to the different layers of HPFM L6 would be $U_{NO3\text{-}HZ15} = 567$ mg $NO_3^-$-N m$^{-2}$ d$^{-1}$ in the shallow layer (0 to 15 cm depths), $U_{NO3\text{-}HZ30} = 174$ mg $NO_3^-$-N m$^{-2}$ d$^{-1}$ in the layer from 15 to 30 cm depths and $U_{NO3\text{-}HZ45} = 256$ mg $NO_3^-$-N m$^{-2}$ d$^{-1}$ in the deepest layer from 30 to 45 cm depths. From this example one could conclude that the shallowest sediments are the most efficient ones in term of nitrate removal. While removal activity is first declining with depths it later increases again. This finding is consistent with the higher amplitudes of oxygen concentration in 45cm depths compared to 25 cm depths, also suggesting higher biotic activity at the deepest layer. Potential reasons for this pattern could be decreasing $NO_3^-$ penetration with depth (lower uptake at the middle layer than the shallowest one) which is in the deepest parts counter balanced by increased residence time and stronger reducing conditions.

# 5. Conclusion and Outlook

The role of the hyporheic zone as a hotspot for instream nutrient cycling is indisputable (Mulholland et al., 1997; Fellows et al., 2001; Fischer et al., 2005; Rode et al., 2015). Quantitative and qualitative knowledge about the influence of mass transfer on hyporheic nutrient removal is crucial to manage streams and rivers, especially in the

light of increasing worldwide morphological alterations (Borchardt and Pusch, 2009), eutrophication (Ingendahl et al., 2009) and sediment loading (Hartwig and Borchardt, 2015). Despite decades of research on hyporheic nutrient cycling, robust quantitative data on nutrient fluxes through the hyporheic zone are limited, which is mainly due to methodological constraints in measuring nutrient concentrations and water flux in the subsurface of streams

(O'Connor et al., 2010; Boano et al., 2014; Gonzalez-Pinzon et al., 2015).Passive flux meters have the potential to fill the gap in measured quantitative nutrient fluxes to the reactive sites in the sediments of rivers. Up to date, HPFM are virtually the only method which can simultaneously capture nutrient and water flux through hyporheic zone within the same device and at the same spatial location. The field testing of several devices proved the general applicability of passive flux meters for quantifying $NO_3^-$ and $PO_4^-$ flux to reactive sites in the hyporheic zone. The

hyporheic flux rates of nutrients and nitrate uptake rates measured in an agricultural $3^{rd}$ order stream were generally in agreement with rates reported in the literature. Our results clearly highlight the advantages of HPFM compared to commonly used methods (i.e. grab sampling of pore water and separate measurements of hyporheic exchange and Darcy velocities), first of all the capacity to integrate over longer time periods.

Quantifying nutrient flux to the potentially reactive sites in the hyporheic zone is an essential step to further improve

our process based knowledge on hyporheic nutrient cycling. In the future, long-term measurements of nutrient fluxes as obtained from HPFM can feed into and advance the transport part of nutrient cycling models.

We anticipate further improvement and increased use of passive flux meter approaches in order to advance conceptual models of nutrient cycling in the hyporheic zone. We demonstrated modifications which extended PFM application from groundwater to hyporheic zones. Current limitations related to the potential bias of results due to

biofilm growth on sorbents require further analysis for the identification of more suitable sorbents. While we focused on nutrients, PFM may also be used for a wide range of other substances like contaminants or trace elements.

Being labor efficient and attractive with respect to relatively low costs, numerous HPFM can be efficiently used to cover larger areas and assess the degree of local heterogeneity. Further, neither advanced technology, maintenance,

or power supply are needed which can be extremely advantageous for the use in remote areas or study sites without infrastructure.

*Acknowledgements*

We thank Uwe Kiwel for his technical support during the field work and Andrea Hoff and Christina Hoffmeister

from the analytical department of the UFZ for their assistance in the laboratory experiments. We are also grateful for the fruitful discussions with James Jawitz, Andreas Musolff, Christian Schmidt and Nico Trauth.

# Figures and tables

**Table 1. Resident tracers per liter of aqueous solution and their partitioning characteristics. Retardation factors ($R_d$) for the specific set of tracers and AC used in this study had previously been determined by Annable et al. (2005)**

| Resident tracers | Aqueous concentration (g L$^{-1}$) | Rd |
|---|---|---|
| methanol | 1.2 | 4.9 |
| ethanol | 1.2 | 20 |
| Isopropyl alcohol (IPA) | 2.3 | 109 |
| tert-butyl alcohol (TBA) | 2.3 | 309 |
| 2,4-dimethyl-3-pentanol (DMP) | 1.2 | >1000 |

**Table 2. Selected morphological and hydrological parameters of the testing site for the duration of the testing phase from 04.06.2015-11.06.2015. Ranges are indicated for directly measured parameters, the remaining parameters have been calculated from listed means. HZ= Hyporheic zone**

**Surface water**

| | acronym | unit | mean | range |
|---|---|---|---|---|
| cross sectional area | $A_{SW}$ | m² | 3.41 | |
| depth | $h$ | m | 0.565 | 0.54 - 0.61 |
| width | $w$ | m | 6.03 | 5.57 - 6.29 |
| mean velocity | $v$ | m s$^{-1}$ | 0.097 | |
| discharge | $Q_{SW}$ | m³ s$^{-1}$ | 0.32 | 0.30 - 0.34 |
| $NO_3^-$ concentration | $C_{NO3\ SW}$ | mg $NO_3^-$-N L$^{-1}$ | 2.86 | 2.16 - 3.26 |
| $NO_3^-$ load | $M_{NO3\ SW}$ | mg $NO_3^-$-N s$^{-1}$ | 896 | |
| $PO_4^-$ concentration | $C_{PO4\ SW}$ | mg P L$^{-1}$ | 0.165 | 0.111 - 0.231 |
| $PO_4^-$ load | $M_{PO4\ SW}$ | mg P s$^{-1}$ | 51 | |

**Hyporheic zone upper 50cm**

| | acronym | unit | mean | range |
|---|---|---|---|---|
| Assessed depth of HZ | $h_{HZ}$ | m | 0.5 | |
| cross sectional area of HZ | $A_{HZ}$ | m² | 3.02 | |

**Table 3. Summarized parameters of NO$_3^-$ transport and removal through the upper 50 cm of the hyporheic zone at the test site for the testing phase from 04.06.-11.06.2015. Ranges are indicated for directly measured parameters, the remaining parameters have been calculated from listed means.**

|  | acronym | unit | mean | range |
|---|---|---|---|---|
| water flow through HZ | $Q_{HZ}$ | L s$^{-1}$ | 0.0265 | |
| % of river water entering HZ | $\%Q_{HZ}$ | % | 0.008 | |
| Horizontal Darcy velocity | $q_x$ | cm d$^{-1}$ | 76 | 20 - 116 |
| average NO$_3^-$ concentration in the HZ | $C_{NO3\,HZ}$ | mg NO$_3^-$-N L$^{-1}$ | 1.39 | 0.31 - 2.86 |
| % NO$_3^-$ entering the HZ which is removed | $R_N$ | % | 52 | |
| potential NO$_3^-$ load entering HZ | $M_{HZ\,theory}$ | mg NO$_3^-$-N s$^{-1}$ | 0.08 | |
| NO$_3^-$ load measured in HZ | $M_{HZ\,measured}$ | mg NO$_3^-$-N s$^{-1}$ | 0.037 | |

**Table 4. Benchmark surface water parameters derived from the continuous sensor records from 04.06.-11.06.2015 and 08.10. – 11.10.2015: Temp= temperature, SpC=specific conductivity, O$_2$ =dissolved oxygen**

|  |  | Temp | SpC | pH | O$_2$ | NO$_3^-$ |
|---|---|---|---|---|---|---|
|  |  | °C | µS cm$^{-1}$ | - | mg L$^{-1}$ | mg NO$_3^-$-N L$^{-1}$ |
| 04.-11. June 2015 | mean | 17.81 | 1063 | 8.42 | 9.37 | 2.86 |
|  | STD | 2.57 | 46 | 0.27 | 2.01 | 0.32 |
|  | min | 13.38 | 886 | 7.75 | 6.13 | 2.16 |
|  | max | 23.79 | 1224 | 8.84 | 13.12 | 3.26 |
|  |  |  |  |  |  |  |
| 08.-11. Oct 2015 | mean | 11.22 | 951 | 8.21 | 10.48 | 2.75 |
|  | STD | 2.75 | 59 | 0.10 | 0.91 | 0.28 |
|  | min | 6.02 | 818 | 7.99 | 9.09 | 1.95 |
|  | max | 15.32 | 1056 | 8.44 | 12.44 | 3.40 |

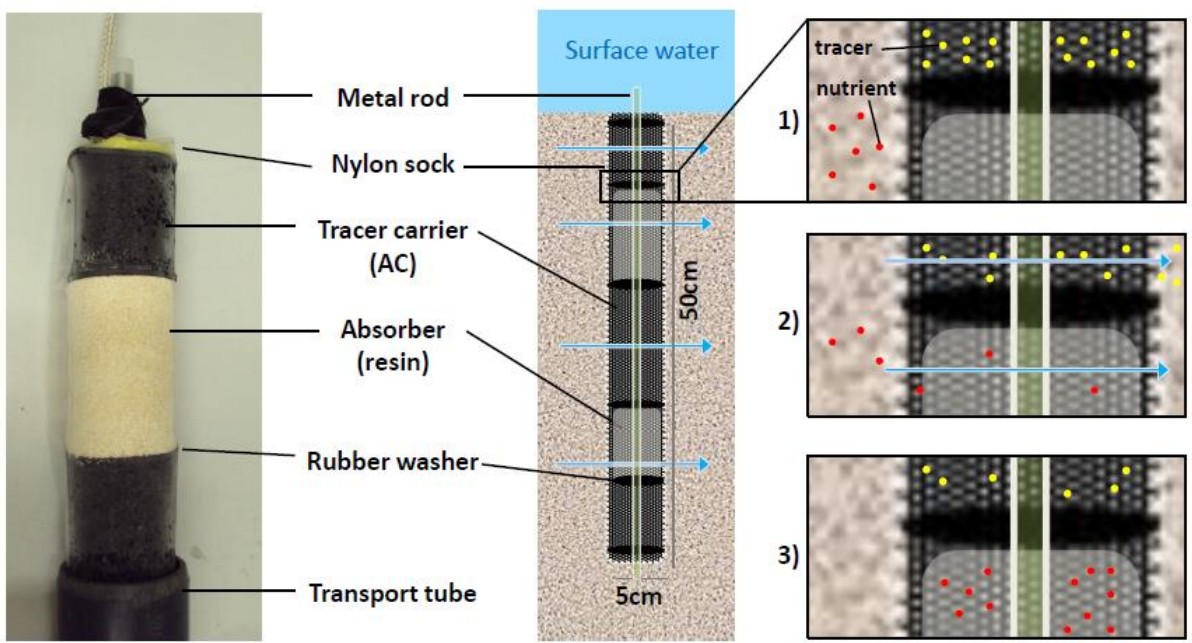

**Figure 1. Photograph of an HPFM with alternating segments before deployment (left) , schematic profile of a deployed HPFM (middle) and schematic steps of HPFM functioning (right): 1) directly after installation, tracer resides on activated carbon (AC), 2) infiltrating water washes out the tracer, nutrients enter the HPFM and are absorbed on the resin, 3) after retrieval nutrients are fixed on the resin, tracer concentration is diluted.**

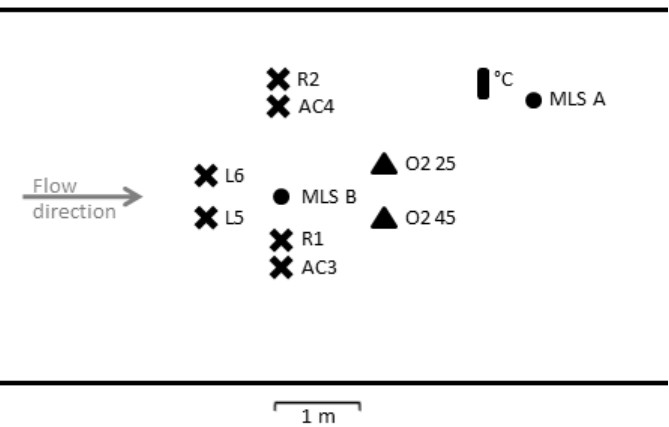

**Figure 2. Overview of the instrumental setup at the Holtemme for the testing phase in June 2015.**
**R1, R2 resin only HPFM; AC3, AC4 activated carbon only HPFM; L5,L6 alternating layered HPFMs; MLSA, MLSB Multi-level sampler; O2 25, O2 45 subsurface oxygen logger; °C vertical temperature profile**

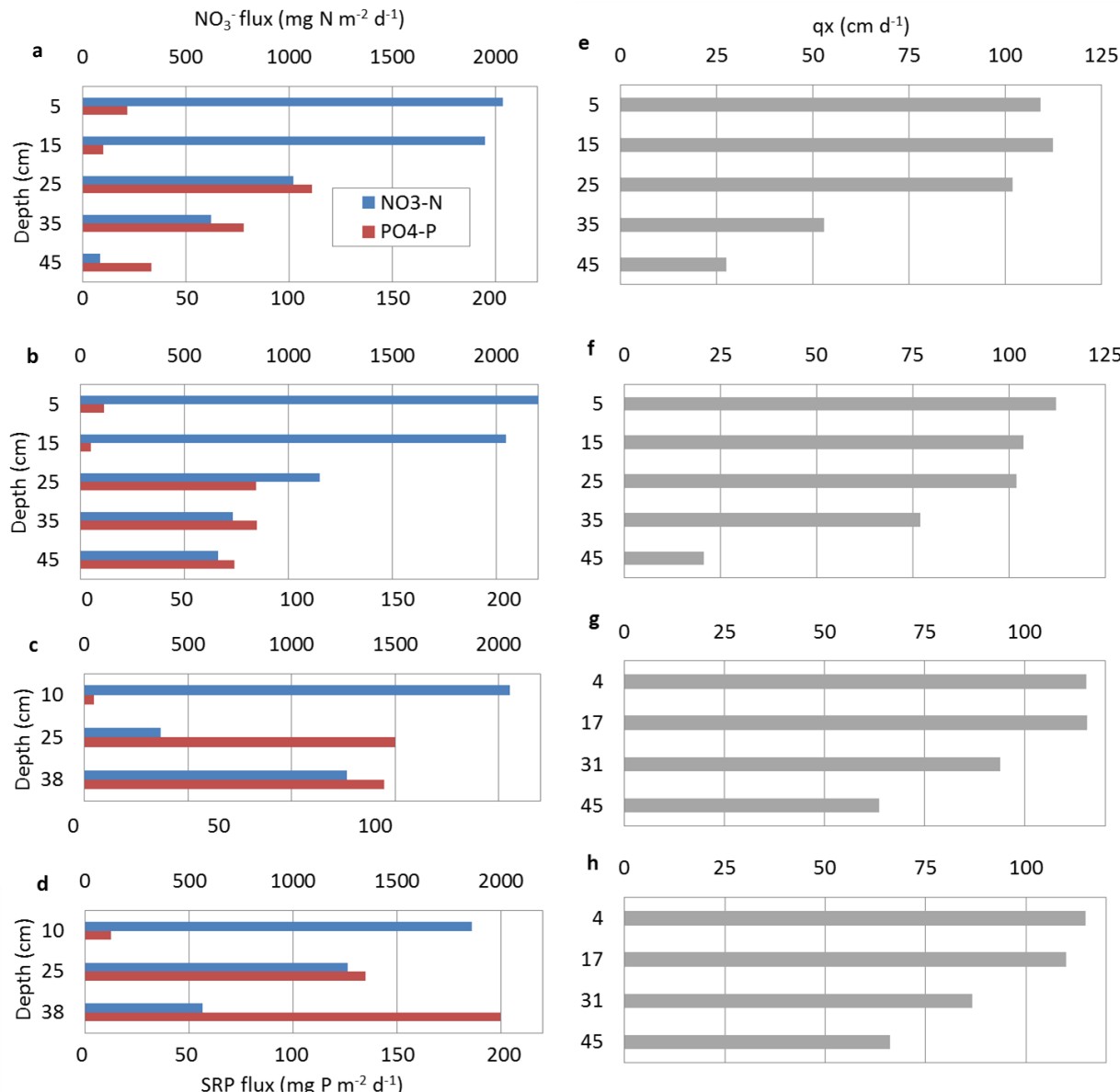

**Figure 3. Time integrative measurements for the 04.-11.06.2015. Left side: Horizontal NO$_3^-$-N and SRP-P flux in mg m$^{-2}$ d$^{-1}$ through the resin HPFM R1 (a), R2 (b) and the layered HPFM L5 (c) and L6 (d). Right side: corresponding Darcy velocities $q_x$ in cm d$^{-1}$ through the activated carbon HPFM AC3 (e) and AC4 (f) and the layered HPFMs 5L (g) and 6L (h)**

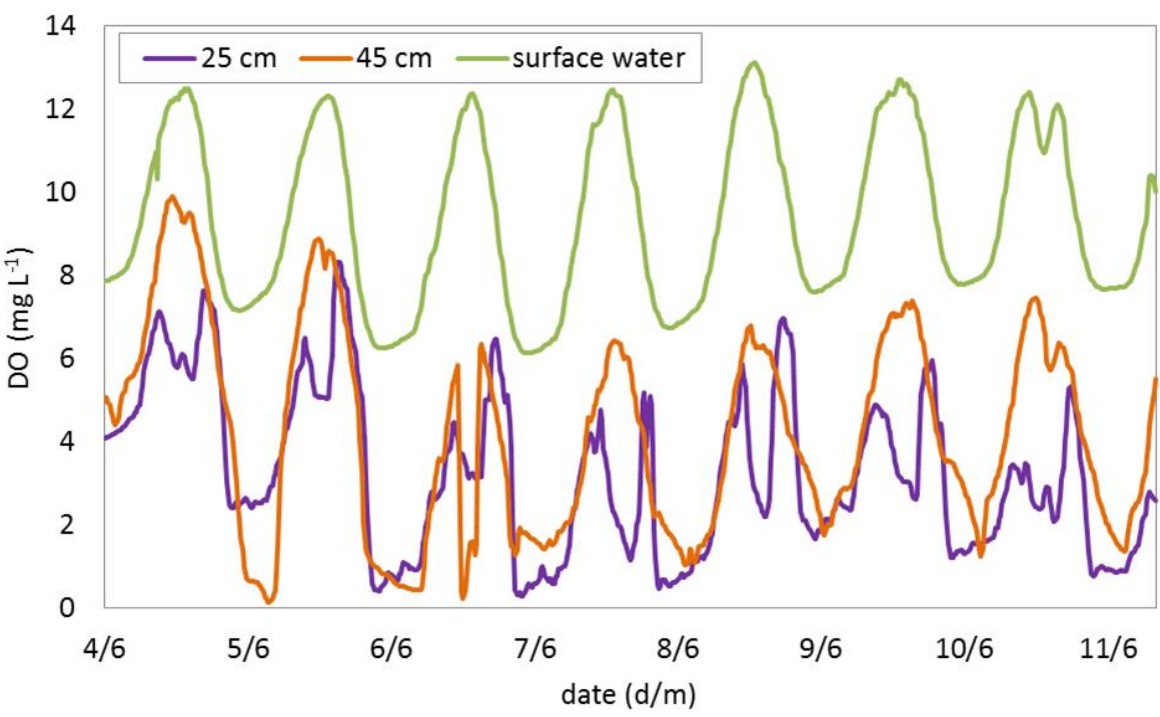

**Figure 4. Time series of dissolved oxygen concentrations in the surface water (green) and the subsurface water (depth 25 cm, purple and depth 45 cm orange) at the study site from 04.-11.06.2015**

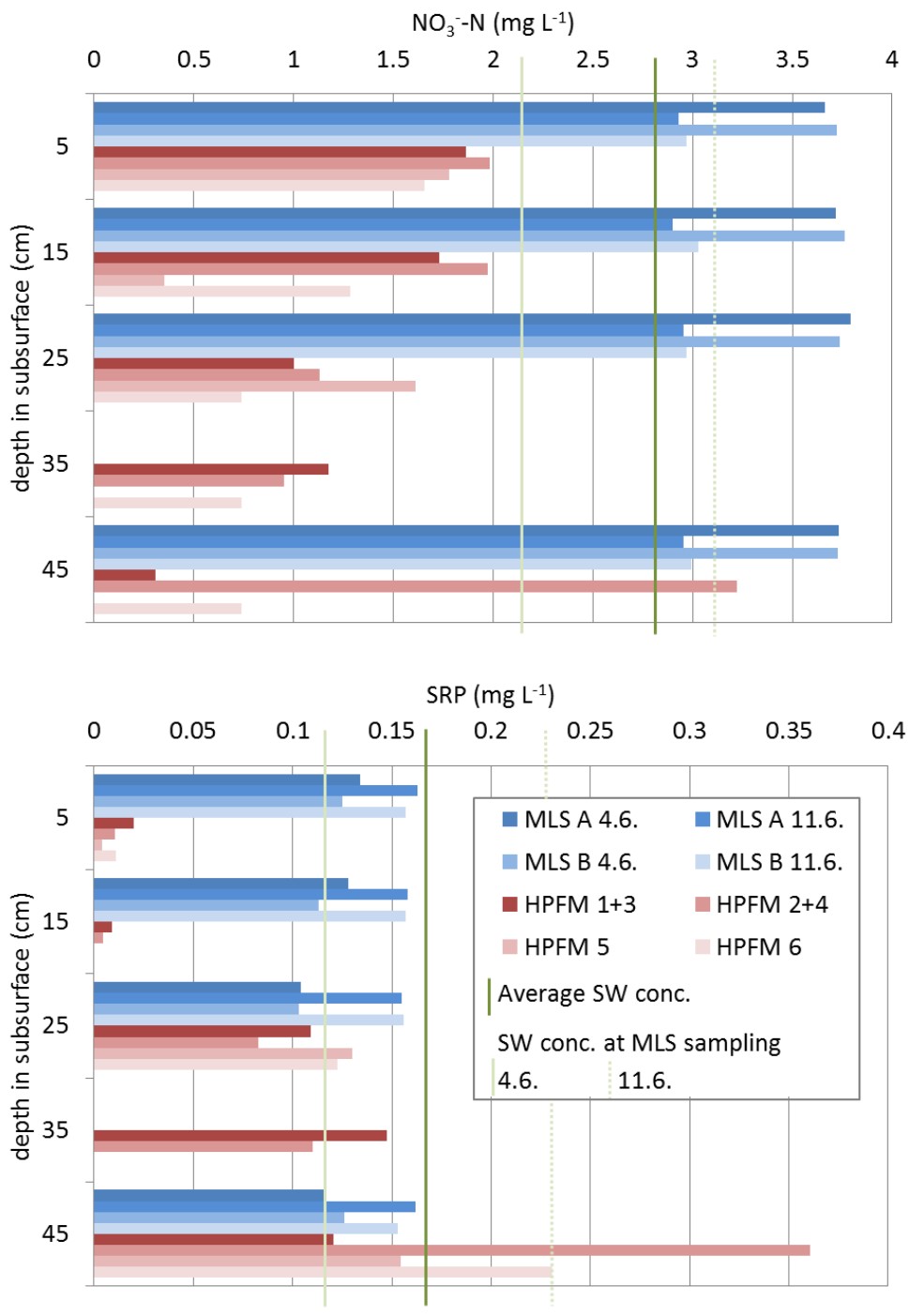

**Figure 5. Comparison between manually sampled pore water from MLS (red) and HPFM (blue) for NO$_3^-$-N (top) and SRP (bottom). Each MLS was sampled on the 04. and 11.06.2015. Average surface water concentration during the deployment time as well as the concentrations at the time point of MLS sampling are marked in green.**

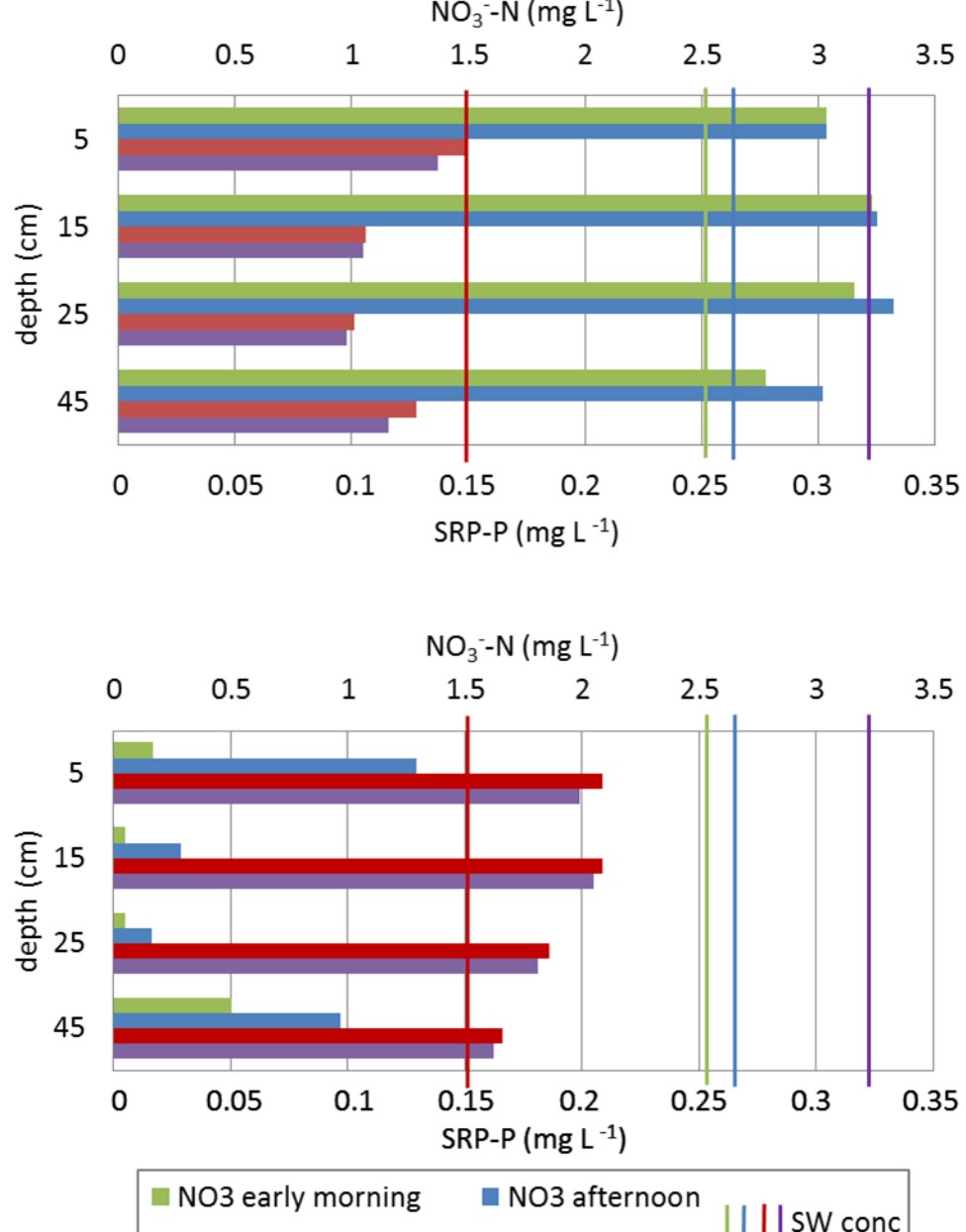

**Figure 6. Concentrations of NO$_3^-$-N and SRP in time differentiating manually taken pore water samples from MLS A (bottom) and MLS B (top) on 8$^{th}$ October 2015. Corresponding surface water concentrations are marked as vertical lines.**

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
