# Peer review of "Quantifying nutrient fluxes in Hyporheic Zones with a new Passive Flux Meter (HPFM)"

_Biogeosciences, 2016_

## Short Comment (SC1) · 24 Aug 2016

The article describes implementing a combination of two passive methods to determine joint water, nitrate, and phosphate flux in a depth profile in a stream bed. Water flow is quantified using a previously published empirical relationship between alcohol dilution from activated carbon, while chemical fluxed employed ion exchange resins. The method was tested both in the field and laboratory.

The work was well described, and appeared to be carried out with attention to many important potential interferences (e.g., attentive to the fact that many brands of ion exchange resin have significant background concentrations of nitrate). In the end the results were believable and encouraging, which is in keeping with the fact that the base methods have been well developed in previous studies. To be clear, the implementation in the hyporheic setting is novel and represents a solid advancement.

The paper was well written, with only a few places where I encountered grammatical usages which I found to be slightly awkward (noted on the attached pdf). With respect to brevity, I found the introduction and discussion to be overly detailed - the role here is to point out the need for this method in the context of previous work. It should be a matter of just one or two sentences to lay out the well-documented importance of hyporheic processes. At the same time, since this is a methods paper, I found that the description of related methods (e.g., heat pulse or passive heat transport for flow) to be lacking specific attribution of relationship to the new method in terms of precision, advantages and disadvantages. thus I found the paper to bely the authors interest in flow processes over methods, which was at odds with the goal of the paper. On this same point, I was expecting that the heat-flow methods described would be reported in the results, which I did not encounter.

Overall, I think this paper is well deserving to be published with minor revisions to attend to the issues I have noted.

John Selker Oregon State University

Please also note the supplement to this comment:
http://www.biogeosciences-discuss.net/bg-2016-334/bg-2016-334-SC1-supplement.pdf

[Figure]

**Supplement:**

[revised manuscript text omitted]

---

## Referee Comment (RC1) · Anonymous Referee #1 · 7 Sep 2016

Content

The article describes a very interesting passive approach to determine and couple interstitial water flow and nutrient transport in the hyporheic zone, the hyporheic passive flux meter (HPFM). The method is based on alcohol dilution from activated carbon and ion exchange resins. They firstly tested the ion exchange resins to obtain the most appropriate one. Secondly tested the HPFM in the field and compared the results with a most commonly used method such as pore water sampling. The presented approach is very interesting since it reduces temporal variability and the sampling effort, and it is very relevant because couples nutrients and water flow, to obtain hyporheic fluxes.

General comments

While interesting and novel, I have five major concerns as summarized herein. First,

[Figure]

I have concerns about the approach in the field test. The number and distribution of the HPFMs in the field seems to be done assuming a very homogeneous hyporheic zone, however this hardly ever happens. As a consequence the high spatial variability arises among all the measurements. There is not enough replication for basic statistical tests, and therefore, the comparison with the reference method, the pore water sampling (MLS), is very difficult to interpret. At least the analysis of the data should be done using those layers that have been measured as replicated presenting means and standard deviations/errors. And when possible perform statistical tests that proof or not the differences.

Second, the data from the HPFM should be explored with more detail and use the information provided by the coupled information to obtain more accurate information. In its present form the usage of the data is slightly superficial. A part of showing whether the HPFMs worked or not, the manuscript should also show which information can be obtained with them.

Third, the growth of biofilm appears to be very significant in the HPFMs both in the laboratory columns and in the field. This could have strongly influenced the results and should be taken into account by the future or potential users of the HPFMs compared with other methods based on diffusion; however no data on this aspect are shown.

Fourth, while the abstract is quite direct, the introduction is too detailed what makes the reading confusing and lacks of a clear and direct objective. The methods section provides numerous and useful details as should be in a methodological manuscript, however they could be arranged in another way more intuitive that eases the reading. In general terms, the manuscript would be more convincing if it were presented as a comparison with MLS.

Fifth, the text is well written in general however a revision of expressions and grammatical mistakes is needed.

For Specific comments see Supplement to this comment.

Please also note the supplement to this comment:
http://www.biogeosciences-discuss.net/bg-2016-334/bg-2016-334-RC1-
supplement.pdf

**Supplement:**

Journal: BG
Title: Quantifying nutrient fluxes in Hyporheic Zones with a new Passive Flux Meter (HPFM)
Author(s): J. V. Kunz et al.
MS No.: bg-2016-334
MS Type: Research article

**Content**

The article describes a very interesting passive approach to determine and couple interstitial water flow and nutrient transport in the hyporheic zone, the hyporheic passive flux meter (HPFM). The method is based on alcohol dilution from activated carbon and ion exchange resins. They firstly tested the ion exchange resins to obtain the most appropriate one. Secondly tested the HPFM in the field and compared the results with a most commonly used method such as pore water sampling. The presented approach is very interesting since it reduces temporal variability and the sampling effort, and it is very relevant because couples nutrients and water flow, to obtain hyporheic fluxes.

**General comments**

While interesting and novel, I have five major concerns as summarized herein. First, I have concerns about the approach in the field test. The number and distribution of the HPFMs in the field seems to be done assuming a very homogeneous hyporheic zone, however this hardly ever happens. As a consequence the high spatial variability arises among all the measurements. There is not enough replication for basic statistical tests, and therefore, the comparison with the reference method, the pore water sampling (MLS), is very difficult to interpret. At least the analysis of the data should be done using those layers that have been measured as replicated presenting means and standard deviations/errors. And when possible perform statistical tests that proof or not the differences.

Second, the data from the HPFM should be explored with more detail and use the information provided by the coupled information to obtain more accurate information. In its present form the usage of the data is slightly superficial. A part of showing whether the HPFMs worked or not, the manuscript should also show which information can be obtained with them.

Third, the growth of biofilm appears to be very significant in the HPFMs both in the laboratory columns and in the field. This could have strongly influenced the results and should be taken into account by the future or potential users of the HPFMs compared with other methods based on diffusion; however no data on this aspect are shown.

Fourth, while the abstract is quite direct, the introduction is too detailed what makes the reading confusing and lacks of a clear and direct objective. The methods section provides numerous and useful details as should be in a methodological manuscript, however they could be arranged in another way more intuitive that eases the reading. In general terms, the manuscript would be more convincing if it were presented as a comparison with MLS.

Fifth, the text is well written in general however a revision of expressions and grammatical mistakes is needed.

**Specific comments**

Page 1 Lines 25-27: In the manuscript, the HPFMs was placed in the streambed for a week and once recovered provided information on the total flux of nutrients and water during the study period however did not provide any information on temporal variability. In fact, pore water sampling could account for much more temporal variability than the presented approach.

Page 1 Lines 9-33: The abstract or the keywords should provide some more detailed information about the method; more specifically include the use of resins and activated carbon.

Page 1 Line 15: it is not clear the meaning of the term load, does it refer to nutrient concentration?

Page 2 Lines 13-14: The definition of hyporheic zone excluded groundwater, however in line 20 (page 2) there is a reference to the significance of groundwater for nitrogen cycling in the hyporheic zone. I suggest expanding the definition, seeing for example Boulton, A. J.; Findlay, S.; Marmonier, P.; Stanley, E. H.; Valett, H. M., The functional significance of the hyporheic zone in streams and rivers. Annual Review of Ecology and Systematics 1998, 29, 59-81.

Page 2 Line 29: In the discussion, the term hotspot has been used, I suggest keeping the term uniform and use it here as well.

Page 2 Line 35: Does "exchange rates" correspond to water, nutrients or both?

Page 4 Line 32: In the present study biofouling is also not clearly regarded since no control or no data are shown.

Page 5 Lines 2-13: Please include information about the resin and AC such as pore size, specific surface, porosity... Please include also an estimation of the maximum potential adsorption, how much mass of the studied nutrient can be measured? Which is the detection limit of the HPFMs for nutrient and which is the minimum Darcy velocity that can be detected?
Please indicate in this section as well that resin and AC aimed to inform about two different parameters/processes.

Page 5 Lines 11-12: Why was this mesh size selected? Taking into account the characteristics of the streambed sediment, of the resin and/or AC, or both? Very fine streambed sediment could clog the mesh, or even enter the resin and AC and clog it. On the other hand, in a very permeable streambed the HPFM would probably act as an impermeable layer and limit the exchange of tracer and nutrients to diffusion. See Ward, A. S., et al. (2011). "How can subsurface modifications to hydraulic conductivity be designed as stream restoration structures? Analysis of Vaux's conceptual models to enhance hyporheic exchange." Water Resources Research 47: W08512.

Page 5 Line 4: Please indicate that the tracer loaded carrier is the activated carbon (AC).

Page 5 Lines 34-35, Page 6 lines 1-6: Were the HPFMs stored dry? When placing the HPFMs in the streambed there was a first wash of the resin and AC, could it be estimated how much is this first contact with stream water influencing the final result? How much of the maximum potential adsorption/dilution (%) is lost in this first step?

Page 6 Line 14: The heading of this section is confusing since it is commonly placed at the end of the methods section; however this is a methodological manuscript. This section would be better merged with the correspondent method, Section 2.2.1 included right after the description of the AC, and section 2.2.2. merged with the description of the resins.

Page 7 Lines 4-6: Is "$J_N$" time-averaged advective horizontal nutrient flux? Please indicate it together with the correspondent units.

Page 7 Line 10: The heading of section 2.3. is a bit confusing, does not reflect the aim of the section, to ease the reading, it might be better to swift this section to right after section 2.1.1.
Page 7 Line 11: If as indicated experiments described in that paragraph were accomplished on triplicate; please present the data as means +/- standard error, or standard deviation.

Page 7 Lines 16-29: Please, provide more details on the experimental setup, for instance were the columns pump bottom-top or top-bottom direction, where the columns placed vertically or horizontally, for how long were the tests run. Please provide the brand of the pump.

Page 7 Line 19: I wonder whether at the same nutrient flux into the HPFMs (high concentrations and low flow, or low concentrations at high flow) the differences in interstitial velocity (i.e. Darcy velocity) would influence the adsorption/dilution due to turbulent flow and finer diffusive boundary layer.

Page 7 Lines 30-32: Please provide these results.

Page 8 Lines 1-2: The lower or higher concentrations of N and P respectively contained after the incubation were used to correct the obtained data in the field experiment?

Page 8 Line 17: Does "stones" refer to Boulders or cobbles? Please specify. If possible, please provide information on the granulometry of the streambed.

Page 8 Line 33 Which is the reasoning for doing such combination of resin and AC? Would the results be more accurate in this way? If the aim is to test simultaneously both approaches, this arrangement does not seem appropriate since each layer is considered independent from the other one, and it is not clearly assumed that the streambed will have uniform nutrient concentrations or interstitial flows.

Page 9 Lines 5-7: Are the presented results corrected with this control?

Page 9 Lines 19-20: it is not clear why the oxygen loggers had to be placed four weeks in advance for re-equilibration, while the HPFMs where placed without re-requilibration period. Would it be wise for future measurements to leave for example a perforated metal case in the streambed for certain period before placing the HPFMs? In this way, would the hyporheic zone be re-equilibrated after hammering the metal case in the streambed?

Page 9 Lines 21-37 and Page 10 Lines 1-3: The manuscript aims to compare the HPFMs with the pore water sampler (MLS), therefore this has to be clearly stated and well explained in the methods.

Page 9 Line 23 and figure 2: Why is the MLS A located so far (>2m) from the rest of the measurement points?

Page 10 Lines 1-3: Are the N and P data from June presented in the manuscript? Or for June just information on $SO_3$ and B are provided? And data from N and P just correspond to October?

Page 10 Lines 4-13: The measurements presented here should have an appropriate heading as the MLS, oxygen profiles… or do these methods belong to the MLS?

Page 10 Line 7: Within the context of the manuscript it is also interesting to provide the detection limit.

Page 10 Lines 10-13: Most of the parameters measured with the YSI probes are not provided in the results or tables. Please include them in a table, with mean and standard error or deviation for the incubation period.

Page 10 Lines 12-13: Which is the relevance of Chlorophyll-a for the aim of the manuscript?

Page 10 Lines 17-19: Indicate clearer, if correct, the abbreviations of all terms: "the proportion of surface water ($Q_{SW}$, $m^3 s^{-1}$) infiltrating…"

Page 10 Lines 17-19: Considering the interesting information about Darcy velocities in the hyporheic zone provided by the HPFM, it would be more accurate to calculate the proportion of infiltrated surface water from the cumulative $Q_{HZ}$ for each layer, so the ratio will be $\Delta Q_{HZ}/Q_{SW.}$
Since one of the advantages of the HPFM is that it can measure nutrients and Darcy velocities simultaneously and at different depths, the results will be more complete using an approach that includes that information.

Page 10 Line 18: I am not sure if the measured velocity in the HPFMs can be described as horizontal it could have also been diagonal. Of course according to the calculation it is horizontal but it confuses the reading especially when the results from the temperature show that there was a very strong vertical downwelling. Another term such as interstitial velocity may be more appropriate.

Page 10 Lines 20-21: Due to the lag between the water entering the hyporheic zone from the surface and the measurements in deeper layers, it is no easy to calculate the removal of any nutrient in the hyporheic zone. The N removal activity of the hyporheic zone, as calculated, seems to underestimate the capacity of the hyporheic zone. Considering the interesting information from each layer provided by the HPFMs, it would be interesting to take advantage of it and calculate the removal as something like what follows:
Since there is a quite strong vertical flow, we can assume that the concentration in one layer depends on the previous one. In this way, it can be calculated that the uptake at each layer results from the difference in fluxes ($layer_y$ – $layer_{y+1}$). Of course, as indicated in the introduction, one has to take into account both flow and concentration, that is why it seems better to use the fluxes and not the concentrations. The combined uptake of all layers will provide the total uptake rate in the studied section of the hyporheic zone that can be then compared with the N flux from the surface water.
Additionally, it could be calculated the amount of N removed in the hyporheic zone to the flux of N in the stream to have larger scale information. This would answer the question of how much does the hyporheic zone removes from what is in the stream/ecosystem?

Page 10 Line 31: How could this influence the results? Please include some data.

Page 11 Lines 33-34: See comment on page 5 lines 11-12, could this observation explain, at least partially, the measured darcy velocities (figure 3) in the deeper layers?

Page 12 Lines 4-7: The high variability between both measurements (A and B), the lack of replicates and hence the lack of statistical tests, makes it difficult to draw such conclusions out of the presented data. The presented values represent very high spatial variability in the hyporheic zone, either due to heterogeneous flow or the presence of hotspot/moments during the day. A more cautious sentence should be used, and in the discussion refer to data from the literature.

Page 12 Lines 8-9: Are these values means or just punctual measurements, standard deviations should be then provided. If enough data are available, a simple statistical test should be applied, for instance, one-way analysis of variance (ANOVA).

Page 12 Lines 13-14: The temperature profiles provided information on vertical downwelling from the surface water. However, the darcy velocity obtained from the HPFMs was named as horizontal, however it is not possible to know which the actual direction of the water was. To avoid confusion with the fact that the flow was strongly vertical it might be better to name the flux obtained in the HPFMs as interstitial velocity and assume the sediment was isotropic.

Page 12 Lines 19-23: Considering the high variability shown with two HPFMs in the manuscript, at least three per parameter should be placed in the streambed (flow or nutrients). Even in channelized rivers, small scale variability and heterogeneous residence time distribution occurs (see for example data from vertical water flux Mendoza-Lera, C. and M. Mutz (2013). "Microbial activity and sediment disturbance modulate the vertical water flux in sandy sediments." Freshwater Science 32(1): 26-35.). Additionally, even if low variability is assumed, such approach would be statistically more consistent.

Page 12 Lines 30-31: Data on substantial biofilm growth are not provided, please include.

Page 12 Lines 31-34: Not only biofouling could influence the results, what about uptake/release of nutrients by the biofilm? Right after placing the device in the streambed in was a sterile substrate, which informed about the nutrient concentration in the water flowing through it, and therefore about the surrounding conditions in the hyporheic zone. However, after certain time the HPFMs become an actual physical substrate where the microbial community developed, and therefore the HPFMs became part of the hyporheic zone. Therefore, the information provided by the HPFMs after the incubation also refers to the community inhabiting it.

Page 13 Lines 4-6: Please provide example of the intrusive measurements of hyporheic flow. As occurs when placing piezometers of smaller diameter than the HPFMs+metal case disturbance is created and likely after removing the metal casing fine sediment was sucked into the HPFMs as happens when placing piezometers. When the HPFMs were removed from the streambed and the measurements perform, was there evidence of fine sediment intrusion? Was there any evidence of clogging in the mesh?

Page 13 Lines 11-17: It would be interesting to know which is the time scale detected by the method. If oscillation occurs within 12 hours or less, could the method be further adapted? For example placing the HPFMs for few hours?

Page 13 Lines 30-32: I am not sure whether such assertions can be done in the light of the limited replication and statistical tests. Since the difference in P concentration has not been proofed (no

statistics) it is no possible to link the dynamics of phosphorous with oxygen. However, it could be interesting to take advantage of the oxygen profiles and determine if the dynamics observed in N and P in the HPFM and/or MLS correlate with the mean oxygen concentration.

Page 13 Lines 36-37: The HPFMs provide information of a flowpath, the length, velocity and residence time of the water before reaching the HPFMs is not known (further tracer tests could be implemented in combination with the HPFMs). Then it is difficult to determine whether there are hot spots for denitrification or hot moments, or both (see Abbott, B. W., et al. (2016). "Using multi-tracer inference to move beyond single-catchment ecohydrology." Earth-Science Reviews). Additionally, downwelling into the hyporheic zone does not occur as a front but rather as semicircular flowpaths, see for instance:
Thibodeaux, L. J. and J. D. Boyle (1987). "Bedform-Generated Convective-Transport in Bottom Sediment." Nature 325(6102): 341-343.
Salehin, M., et al. (2004). "Hyporheic exchange with heterogeneous streambeds: laboratory experiments and modeling." Water Resource Research 40: W11504.
Rehg, K. J., et al. (2005). "Effects of suspended sediment characteristics and bed sediment transport on streambed clogging." Hydrological Processes 19(2): 413-427.\

Page 14 Line 10: I suggest including in this section an summary in the form of a table of the factors that should be taken into account for applying the HPFMs in different contexts, for example pH should be taken into account in an acidic stream in a mining area, or permeability is to be taken into account to approximate the permeability of the studied reach (for instance the following method could be used to define the appropriate permeability in the HPFMs Datry, T., et al. (2014). "Estimation of sediment hydraulic conductivity in river reaches and its potential use to evaluate streambed clogging." River Research and Applications 31(7): 880–891).

Page 14 Lines 2-5: I agree, the hyporheic zone is probably deeper than the scale of the HPFMs, however it is as well very heterogeneous in residence time distribution and therefore increasing the number of measurements and/or the scale will be a very interesting. This, together with the high variability of the provided data, indicates that future applications of HPFMs should have enough replicates.

Page 15 Line 20: When more than one author it is uncommon to acknowledge in first person.

Table 2: Please include either ranges or means +/- standard deviation.

Table 3: Please include either ranges or means +/- standard deviation.

Figure 1: Please indicate to which approach corresponds the picture resin or AC, or alternating segments.

Figure 2: For a non-german reader this map won't be very informative, especially because the aim of the manuscript is not to study that stream. I encourage presenting other information instead such as the temperature profiles.

Figure 3: It might be more appropriate to express the fluxes per volume of sediment, in this way confusion with the denitrification rate units would be avoided. Additionally, in table 2, mass fluxes for nutrients are expressed with other units.

Figure 5: I am not familiar with the symbol for diameter to indicate mean concentrations, and it is a bit confusing. I suggest to simply adding surface water.
Considering the variability of the values measured in the hyporheic zone, it would be interesting to include the range of concentrations, or mean +/- standard error or deviation, of the surface water during the deployment time, as understood from the method the nutrient concentrations were measured every 15 mins.

**Technical corrections**

Page 2 Line 18: Correct phosphate, per phosphorous, or P per $PO_4^-$

Page 3 Line 20: Correct technics, per techniques.

Page 4 Line 34: Space missing in "…Pin…"

Page 7 Line 35: Correct KCL per KCl

Page 8 Line 13: Correct figure 2 instead of figure 3

Page 10 Line 2: Correct sun rise, per sunrise

Page 10 Line 20: Correct $NO_3^-$, per $NO_3^-$-N

Page 12 Line 4: Correct SRP, per SRP-P

Page 12 Line 14: For which sentence stands the citation (Layton, 2015)?

Page 14 Line 20: Correct hot spot, per hotspot

---

## Author Comment (AC1) · 13 Sep 2016

Dear John Selker, thank you very much for the comments to our article!

Your comment encouraged us to revise our manuscript. Please see the updated version in the supplement of this response.

Also see below the remarks to your questions/comments.

We admit that the temperature profiling was not adequately accounted for in our article. In response to your remark on p 12, line 9 ("I seem to have missed the comparison to the heat/temperature profile method"): Direct comparison between fluxes detected in the HPFM and those derived from temperature profiles is not possible, because the temperature profile assesses vertical (here dominantly downwards) flow while the

HPFM assess HORIZONTAL flow. We provide a quotient of both from which we calculate an angle of hyporheic flow. Vertical flux was measured in this study in order to supplement our measurements, not as a comparative measurement of flux assessed in the HPFM. We tried to conduct the heat pulse method describe for example by Lewandowski et al. [2011], which can be used to measure small scale hyporheic flow velocities and directions. This method worked well to investigate small scale spatial heterogeneity of hyporheic flow in the shallow (< 10cm) layer of sandy stream beds. However, the gravel at our study site was too coarse, so that installation of the instrument was not practicable. Lewandowski, J., L. Angermann, G. Nützmann, and J. H. Fleckenstein (2011), A heat pulses technique for the determination of small-scale flow directions and flow velocities in the streambed pof sand-bed streams, Hydrological processes, 25.

Page and lines statements in the following text are referring to the revised article! We added several lines on temperature profiling as used in this study to the intro, also highlighting the differences to the Darcy fluxes derived from HPFM measurements. p.4, line 2ff High resolution vertical temperature profiles have efficiently been used to derive vertical Darcy velocity (qy) (m d-1) in the streambed. This methods is based on time series measurements of temperature in the stream and in the sediments at several depths. Based on a numerical model vertical flow velocities can then be calculated from the measured attenuation and phase shift of the diurnal temperature signal which, at depth, varies with the vertical hyporheic flux [Keery et al., 2007; Schmidt et al., 2014] While those vertical Darcy velocities measurements are a valuable supplement, horizontal fluxes are also needed in order to assess hyporheic transport and residence time.

Likewise, the section describing the method in the method-parts was supplemented. p.9, line 29 A numerical solution of the heat flow equation was then used in conjunction with Dynamic Harmonic Regression signal processing techniques for the analysis of these temperature time series. The coded model was kindly provided by the authors

of the above mentioned study. We used vertical Darcy velocities qy to supplement horizontal fluxes assessed with the HPFM in order to estimate the general direction of flow (upwards or downwards) and to calculate the angle of hyporheic flow.

A sentence was added to the results, underlining the differences between qx and qy p12, line 3 With this, vertical flow qy was slightly lower than average horizontal flow qx.

And finally we discussed the usage of temperature profiling in hyporheic studies p14, line 24 Highly resolved temperature profiles enabled us to derive vertical velocities of hyporheic zone. Temperature loggers deliver continuous data, which allow estimating vertical Darcy flux for longer time periods. We therefore consider this method an ideal supplement to the horizontal fluxes assessed with the HPFM. At our study site, vertical water movement was constantly downward and the lowest concentrations of NO3- were observed in the deepest segments of the HPFMs, thus the hyporheic zone at this study site likely extends deeper than the 50 cm evaluated. Addressing your further remarks, we now listed all citation chronologically and corrected the grammatical errors you noted. We added a few lines to the introduction emphasizing the importance of the hyporheic zone. p. 2, line 15 Hyporheic zone processes can substantially modify surface water chemistry during propagation through river networks [Harvey et al., 2013; Boano et al., 2014]. Naegeli and Uehlinger [1997] estimated that the hyporheic zone contributed between 76 and 96% to whole ecosystem respiration. Other studies documented a wider range of hyporheic contribution to whole stream respiration, e.g. 40 to 93 % [Fellows et al., 2001], but agreeing on the importance of hyporheic zone processes on overall stream metabolism. Responding your comment p10, L31 concerning the observed biofilm growth on resin (p11, line 15) "Can this not be eliminated by using a silver treatment as a fixed surface biocide?"

Yes, but it would drastically change the absorbing capacity of the MACROPOROUS resin granulars. Testing different biocides or sterile resins is complex, as for each the complete characterization of the resin behavior has to be repeated and interference with alcohol and nutrient analytics has to be excluded. For this initial

study to implement the method, we only assessed the potential importance of biofilm. As outlined in the discussion, ongoing work with HPFM should also tackle this problem.

Please also note the supplement to this comment:
http://www.biogeosciences-discuss.net/bg-2016-334/bg-2016-334-AC1-supplement.pdf

—————————————————————

[Figure]

**Supplement:**

**Quantifying nutrient fluxes in Hyporheic Zones with a new Passive Flux Meter (HPFM)**

Julia Vanessa Kunz[1*], Michael Annable[2], Jaehyun Cho[2], Wolf von Tümpling[1], Kirk Hatfield[2], Suresh Rao[3], Dietrich Borchardt[1], Michael Rode[1]

[1]Helmholtz Centre for Environmental Research UFZ, Magdeburg, Germany

[2]University of Florida, Gainesville, Florida (USA)

[3]Purdue University, Lafayette, Indiana (USA)

*Correspondence to:* Julia Vanessa Kunz (vanessa.kunz@ufz.de)

**Abstract.** The hyporheic zone is a hotspot of biogeochemical turnover and nutrient removal in running waters. However, due to methodological constraints, our quantitative knowledge on nutrient fluxes to those reactive zones is still limited.

In groundwater systems passive flux meters, devices which simultaneously detect water and nutrient flows through a screen well in the subsurface, proofed to be valuable tools for load estimates.

Here we present adaptations to this methodology and a smart deployment procedure which allow its use for investigating water and solute fluxes in river sediments. The new hyporheic passive flux meter (HPFM) delivers time integrating values of horizontal hyporheic nutrient fluxes for periods of several days up to weeks. Especially in highly heterogeneous environments like the hyporheic zone, measuring flow and nutrient concentration in a single device is preferable when compared to methods that derive flux estimates from separate measurements of water flows and chemical compounds.

We constructed HPFMs of 50 cm length, separated in 5-7 segments which allowed for vertical resolution of horizontal nutrient and water transport in the range of 10 cm. The results of a seven day long field test, which included simultaneous measurements of oxygen and temperature profiles and manual sampling of pore water, revealed further advantages of the method: While grab sampling of pore water could not account for the high temporal variability of nitrate fluxes in our study reach, HPFMs accumulatively captured reliable values for the complete deployment time. Mass balances showed that more than 50 % of the nitrate entering the hyporheic zone was removed in the assessed area.

Being low in costs and labor effective, many flux meters can be installed in order to capture larger areas of river beds. The extended application of passive flux meters in hyporheic studies has therefore the potential to deliver the urgently needed quantitative data which is required to feed into realistic models and lead to a better understanding of nutrient cycling in the hyporheic zone.

Keywords: hyporheic exchange, nutrient retention, quantitative methods, running waters, stream metabolism

**1 Introduction**

Northern and central European rivers export high loads of nitrogen from inland catchments to the marine environment. The ecological and economic problems caused by eutrophication of coastal and riverine ecosystems systems have been recognized years ago (Patsch and Radach, 1997; Artioli et al., 2008;Skogen et al., 2014;). Decades of nutrient studies have unveiled, that rivers cycle rather than only transport nutrients (Garcia-Ruiz et al., 1998a; Seitzinger et al., 2002;Galloway et al., 2003;). However, the quantitative dimensions of instream dynamics of nitrogen (N) and other nutrients are still not completely understood (Wollheim et al., 2008;Grant et al., 2014).

Even though dissimilatory nitrate reduction to ammonium (DNRA) and subsequent anaerobic ammonium oxidation (anammox) might be of importance in some systems (Smith et al., 2015), in most river systems, nitrate ($NO_3^-$) removal via denitrification, the anaerobic reduction of $NO_3^-$ to gaseous $N_2$ is the dominant dissimilatory process which removes N out of the system (Laursen and Seitzinger, 2002;Bernot and Dodds, 2005;Lansdown et al., 2012). Various studies found that in-stream denitrification exclusively happens at "reactive sites" in the hyporheic zone (Duff and Triska, 1990;Rode et al., 2015). The hyporheic zone is defined as the subsurface region of streams and rivers that exchanges water, solutes and particles with the surface (Valett et al., 1993). Hyporheic zone processes can substantially modify surface water chemistry during propagation through river networks (*Harvey et al.*, 2013 *Boano et al., 2014*). *Naegeli and Uehlinger* (1997) estimated that the hyporheic zone contributed between 76 and 96% to whole ecosystem respiration. Other studies documented a wider range of hyporheic contribution to whole stream respiration, e.g. 40 to 93 % (*Fellows et al.*, 2001), but agreeing on the importance of hyporheic zone processes on overall stream metabolism. Consistently, 
[revised manuscript text omitted]

---

## Referee Comment (RC2) · R. González-Pinzón (Referee) · 20 Sep 2016

[referee-annotated manuscript omitted]

---

## Referee Comment (RC3) · J. Lewandowski (Referee) · 20 Sep 2016

Vanessa Kunz and coworker present in the manuscript "Quantifying nutrient fluxes in Hyporheic Zones with a new Passive Flux Meter (HPFM)" a novel technique to measure horizontal water fluxes and nutrient fluxes in hyporheic zones. Without doubt this is an exciting technique to answer unsolved questions about transport and turnover in hyporheic zones. Up to now the lack of adequate techniques hindered in-depth investigations of transport and turnover in this important transition zone.

Major comment

My major concern about this method is its impact on subsurface flow paths and flow velocities. As a consequence the calculated loads might be misleading. Since the device will be placed in sediments with different hydraulic conductivities, its hydraulic conduc-

tivity will sometimes be larger than that of the rest of the sediment and sometimes smaller. In cases where the hydraulic conductivity is lower than that of the surrounding sediment most flow paths will bend around the device instead of passing it. As a consequence there will be much less uptake of nutrients and smaller flow rates. In the opposite case flow will be "sucked" into the device. Nutrient uptake and flow rates will be overestimated. This question could have been addressed in a lab experiment with a box filled with different sediment types with known horizontal flow and a HPFM placed in the center of the box. Alternatively, modelling would also be a method to address this problem. Even without additional investigations it is necessary to carefully discuss this shortcoming and even mention it in the abstract.

Minor comments

In general the paper is very well written and I could only identify a few typing and grammatical errors. The only section of less quality is the abstract. I had the feeling that this was written in a hurry after finalizing the rest of the manuscript. However, as most central part of the manuscript it deserves more care to assure the high quality of the rest of the manuscript. The introduction is well written but relatively long. You might consider to slightly shorten it. The material and methods section is also relatively long and sometimes a bit confusing. Consider to improve its structural elements. The results section is short. The discussion is very well written and of optimum length. The same applies to the conclusions section.

The left part of Fig. 2 can be removed. I do not see any need for this figure and it is so small that it is impossible to see anything here. I recommend refering to another paper where such a map has been included instead of this one here.

P2L24ff: "In stagnant waters, such as lakes, the transport of dissolved nutrients to the sediments is dominantly controlled by diffusion. Therefore, surface water concentrations of nutrients are a good predictor for uptake processes and potential limitations (Dillon and Rigler, 1974;Jones and Bachmann, 1976)." Both sentences are completely

wrong. Diffusion is only a relevant transport process over very short distances but not in a water body or as transport process from a lake to its sediment. Diffusion is a relevant process in the diffusive boundary layer above the sediment surface. In the water body there are many active transport processes such as wind- and temperature-induced transport. You can also discuss this with the people at UFZ Magdeburg involved in lake physics. Transport of nutrients to the sediment occurs mainly in particulate (and not in dissolved) form. In lake sediments many different advective transport processes occur in addition to diffusion, for example groundwater discharge, wave- or seiches-induced pore water transport in the sediment and bioturbation. The latter is especially relevant in shallow lakes. For example, for Lake Müggelsee in Berlin it is well-known that chironomids pump the entire water body through the sediment once a week. Referring to the second of the above cited sentences: Nutrient concentrations in surface waters of lakes are mainly controlled by processes in the water column. For example SRP concentrations are controlled by the very efficient uptake of SRP by plankton during the growing season. That is the reason why SRP concentrations even in eutrophic lakes are usually low during the summer.

In the introduction I would expect a paragraph about hyporheic flow and methods to measure hyporheic flow. The HPFM aims on measuring water and nutrient fluxes but the introduction solely focusses on nutrient fluxes. Please add something about determination of hyporheic flow. For example the heat pulse sensor of Lisa Angermann could be mentioned here but also other methods. I know that the HPS did not work at your site but nevertheless you can mention that there is a device that can be used in finer sediments.

In general I think you consider the sediment only as a sink and not of a source of nutrients. For example in line 11 on page 1 you write "nutrient removal" although the hz is sometimes a source. Besides a temporary storage (e. g. uptake and later release of nutrients) the transport of particulate organic matter to the sediment surface should also be considered. Once this organic material is buried it can release nutrients

**BGD**

as a consequence of mineralization processes. Keep this in mind throughout your manuscript.

The sediment description on page 8 is quite poor. It would be great to know a little bit more about the grain distribution or hydraulic conductivity of the sediment.

P12L19ff "1) In stagnant waters, such as lakes, the transport of dissolved nutrients to the sediments is dominantly controlled by diffusion. Therefore, surface water concentrations of nutrients are a good predictor for uptake processes and potential limitations (Dillon and Rigler, 1974;Jones and Bachmann, 1976)." I do not see the usefulness of this device. I think there is always much more small scale variability in hyporheic zones than assumed. Even in channels. I would recommend placing two alternating HPFMs in the sediment instead and place them with a small depth variation. In that case you will end up with the same spatial resolution but you can assure that flow and nutrient fluxes match to each other. Also, in the entire paper I have not understood the motivation for the separated HPFMs.

Summarizing, I recommend accepting the manuscript after minor revision.

Jörg Lewandowski, IGB Berlin

Please also note the supplement to this comment:
http://www.biogeosciences-discuss.net/bg-2016-334/bg-2016-334-RC3-supplement.pdf

**Supplement:**

[revised manuscript text omitted]

---

## Referee Comment (RC4) · J. Rozemeijer (Referee) · 20 Sep 2016

Review of Kunz et al, Quantifying nutrient fluxes in Hyporheic Zones with a new Passive Flux Meter (HPFM).

The paper presents a useful monitoring tool to increase our understanding of nutrient cycling in the hyporheic zone of streams. The paper is very well organized and written. I only have some minor comments that could be considered by the authors.

P5 l6: From figure 1 the idea of tracer release and nutrient absorption is not immediately clear. Also, the difference between the sections filled with tracer and with absorber is not clear in the graphic (same color)

P6 l32 and P7: "if not indicated otherwise" Remove?

[Figure]

P9l8: At this point in the paper it is not clear why these additional measurements are also done. Consider adding a short introduction.

P10l28: missing r in break

P11l7: Here you mention declining concentrations with depth, but figure 3 shows fluxes. Adding concentrations to figure 3 would also be informative

P11l13: For this conclusion (52% removal), you need to know that the vertical flux is downward and that groundwater has no impact on the concentration levels. However, the vertical flow is given after this conclusion. Re-order?

P12l6: Higher should be lower?

P12l10: Discussion: You may consider to add a paragraph about the applicability of the HPFM. Can it also be applied to quantify vertical nutrient fluxes in lakes and other non-flowing surface waters? Is it applicable in case of a coarse grained hyporheic zone (stones, gravel)?

P12l10: Discussion: The difference in concentrations measured in the MLS are quite different from the HPFM (figure 5). Is this only due to the diurnal variations? Other explanations? How do we know which method is the best one?

P12l14: remove second(Layton , 2015)

P12l18 two points at the end

P14l15: Is this really permanent removal for $PO_4$? Or can it later be released from its absorption sites?

---

## Author Response (AR1)

**Responses to reviewers**

Indicated pages and lines refer to the untracked revised manuscript

**Anonymous reviewer**

**Content**

The article describes a very interesting passive approach to determine and couple interstitial water flow and nutrient transport in the hyporheic zone, the hyporheic passive flux meter (HPFM). The method is based on alcohol dilution from activated carbon and ion exchange resins. They firstly tested the ion exchange resins to obtain the most appropriate one. Secondly tested the HPFM in the field and compared the results with a most commonly used method such as pore water sampling. The presented approach is very interesting since it reduces temporal variability and the sampling effort, and it is very relevant because couples nutrients and water flow, to obtain hyporheic fluxes.

**General comments**

While interesting and novel, I have five major concerns as summarized herein.

First, I have concerns about the approach in the field test. The number and distribution of the HPFMs in the field seems to be done assuming a very homogeneous hyporheic zone, however this hardly ever happens. As a consequence the high spatial variability arises among all the measurements. There is not enough replication for basic statistical tests, and therefore, the comparison with the reference method, the pore water sampling (MLS), is very difficult to interpret. At least the analysis of the data should be done using those layers that have been measured as replicated presenting means and standard deviations/errors. And when possible perform statistical tests that proof or not the differences.

We agree that a much higher number of samplers would be needed to derive representative values. In general and especially for the calculations on uptake rates and denitrification potential we make it more clear in the revised version, that this study is mainly a method development and that the calculated rates are only an example for the use of HPFMs in exploring nutrient dynamics in the stream.

Where possible (e.g. experiments on triplicates, water quality parameters in table 4) we added standard deviation or ranges of values (table 3).

We did not aim to use MLS as a reference method to verify HPFM results, which is not possible because both method measure different things: MLS allow for snap shot sampling of pore-water CONCENTRATION, HPFM deliver average nutrient FLUXES over longer time periods.
We did want to compare the general characteristics of both methods, highlighting the benefits of time integrative measurements.
The revised version focuses more on the method development, also making this point clearer.

Second, the data from the HPFM should be explored with more detail and use the information provided by the coupled information to obtain more accurate information. In its present form the usage of the data is slightly superficial. A part of showing whether the HPFMs worked or not, the manuscript should also show which information can be obtained with them.

As mentioned above, the revised manuscript has a clear focus on the method evaluation. We agree that demonstrating the application of the HPFM and showing options on data treatment should be included. On the same time, it is true that we did not do enough repetitions to make robust statements about our study system. In the revised version we clarify that the field test should serve as an exemplary application in order to test the installation and retrieval procedure and the overall performance of the HPFM. We exemplarily show how the data can be used to interpret hyporheic nutrient dynamics, also adding a depths-wise calculation as suggested. Due to the lack of sufficient data, we don't want to go into too much detail here. We also focused introduction and discussion on the usage of hyporheic nutrient fluxes and why they are important. We think that from this point of view it should be clear, how HPFM data can be used.

Third, the growth of biofilm appears to be very significant in the HPFMs both in the laboratory columns and in the field. This could have strongly influenced the results and should be taken into account by the future or potential users of the HPFMs compared with other methods based on diffusion; however no data on this aspect are shown.

We admit that this topic was not represented adequately. We supply the results of the biofilm experiments as supplements (APPENDIX A). In general, the growth of biofilm on the resin granules and the failure to clearly quantify the (potential) effect of biofouling is the mayor limitation of the method and the major challenge for further improvement of the technique. We clearly state this now in the abstract and conclusion and suggest further steps in solving this problem and improve the performance of HPFMs. Also we added a few lines of discussion about potential effects from biofouling on resin/HPFM. Also see point 42.)

Fourth, while the abstract is quite direct, the introduction is too detailed what makes the reading confusing and lacks of a clear and direct objective. The methods section provides numerous and useful details as should be in a methodological manuscript, however they could be arranged in another way more intuitive that eases the reading. In general terms, the manuscript would be more convincing if it were presented as a comparison with MLS.

We edited the introduction, focusing more on the methodological part and clearly stating hypothesis and aims of the study.
The method section was rearranged. However, as mentioned above, direct comparison between MLS and HPFM measurements does in our opinion not make sense.

Fifth, the text is well written in general however a revision of expressions and grammatical mistakes is needed.

We corrected mistakes marked by you and other reviewers and double checked the revised version

**Specific comments**

1.) Page 1 Lines 25-27: In the manuscript, the HPFMs was placed in the streambed for a week and once recovered provided information on the total flux of nutrients and water during the study period however did not provide any information on temporal variability. In fact, pore water sampling could account for much more temporal variability than the presented approach.

Formulation corrected: "*Due to the high temporal variability of nutrient fluxes in the subsurface of our study reach, single grab samples of pore water could not be used to characterize overall fluxes. With HPFMs accumulative values for the average flux during the complete deployment time could be captured, while on the same time reducing the sampling effort.*" P1 L24ff

2.) Page 1 Lines 9-33: The abstract or the keywords should provide some more detailed information about the method; more specifically include the use of resins and activated carbon.

Both were added to the key words and included in the abstract "*Their functioning is based on accumulation of the substances on a sorbent and concurrent dilution of a resident tracer which is previously loaded on the sorbent.*" P1 L17ff

3.) Page 1 Line 15: it is not clear the meaning of the term load, does it refer to nutrient concentration?

Changed to "*fluxes of target solutes and water through those ecosystems." P1 L17*

4.) Page 2 Lines 13-14: The definition of hyporheic zone excluded groundwater, however in line 20 (page 2) there is a reference to the significance of groundwater for nitrogen cycling in the hyporheic zone. I suggest expanding the definition, seeing for example Boulton, A. J.; Findlay, S.; Marmonier, P.; Stanley, E. H.; Valett, H. M., The functional significance of the hyporheic zone in streams and rivers. Annual Review of Ecology and Systematics 1998, 29, 59-81.

Definition completed to "*The hyporheic zone, the subsurface region of streams and rivers that exchanges water, solutes and particles with the surface (Valett et al., 1993) and may mix stream-water during the transport through the sediments with underlying groundwater (Triska et al., 1989; Fleckenstein et al., 2010;Trauth et al., 2014)" P2 L11*

5.) Page 2 Line 29: In the discussion, the term hotspot has been used, I suggest keeping the term uniform and use it here as well.

We don't see how this word fits into here, or how this paragraph is referring to a hotspot. However, this paragraph was edited and reformulated

6.) Page 2 Line 35: Does "exchange rates" correspond to water, nutrients or both?

Water and solutes as clearly stated in the sentence before. *P3 L2*

7.) Page 4 Line 32: In the present study biofouling is also not clearly regarded since no control or no data are shown.

We reformulated the paragraph, stating that biofouling was considered as a potential challenge in our application. We also extended discussion on biofouling and provide the results from the experiments as supplementary material. Please see my comment above.

8.) Page 5 Lines 2-13: Please include information about the resin and AC such as pore size, specific surface, porosity... Please include also an estimation of the maximum potential adsorption, how much mass of the studied nutrient can be measured? Which is the detection limit of the HPFMs for nutrient and which is the minimum Darcy velocity that can be detected?
Please indicate in this section as well that resin and AC aimed to inform about two different parameters/processes.

Information about resin (P6 L14ff) and AC (P7 L4f) were added to the respective paragraphs. Detection limit for nutrient accumulation on the resin was included in the methods "*The limit of quantification LQ for the nutrient extraction resulting from this background was calculated according to the EPA Norm 1020B (Greenberg et al., 1992) as the sum of background concentration and 10 times the standard deviation and amounted to 24 µg NO3- g-1 resin and 0.097 µg PO4- g-1 resin.*" P6 L9ff

An estimate of an overall detection limit is provided now in the discussion: "*As the values derived from the control incorporate all the processing steps of HPFMs and samples, they can be regarded as the method detection limit MDL (Greenberg et al.,1992). The MDL defines the lower limit for the use of HPFMs in cases were nutrient fluxes are very low and deployment time cannot be extended*" P14 L31ff

We clarify the different aims of resin/AC: "*was filled with a mixture of a macroporous anion exchange resin as a nutrient absorber and alcohol tracer loaded activated carbon (AC) for the water flow quantification.*" P5 L7

9.) Page 5 Lines 11-12: Why was this mesh size selected? Taking into account the characteristics of the streambed sediment, of the resin and/or AC, or both? Very fine streambed sediment could clog the mesh, or even enter the resin and AC and clog it. On the other hand, in a very permeable streambed the HPFM would probably act as an impermeable layer and limit the exchange of tracer and nutrients to diffusion. See Ward, A. S., et al. (2011). "How can subsurface modifications to hydraulic conductivity be designed as stream restoration structures? Analysis of Vaux's conceptual models to enhance hyporheic exchange." Water Resources Research 47: W08512.

We completed the method section on the selection of mesh size
"*In general, meshes should be as wide as possible because very fine mesh may act as a barrier to water flow limiting infiltration of water and solutes into the HPFM (Ward et al., 2011). However, the mesh should be smaller than the finest sediments, AC or resin granules.*" P5 L14f
We did not observe neither clogging nor infiltration of fine particles. We agree that this is an issue which should be regarded and included it into our discussion "*If fine particles are observed to bypass the mesh and enter the HPFM a finer mesh should be chosen. We did not observe clogging of the mesh or intrusion of particles at our study, though in highly permeable systems with fine particle transport this might have to be considered.*" P16 L5ff

10.) Page 5 Line 4: Please indicate that the tracer loaded carrier is the activated carbon (AC).

Completed: "*The Hyporheic Passive Flux Meters (HPFMs) consisted of a nylon mesh which was filled with a mixture of a macroporous anion exchange resin as a nutrient absorber and alcohol tracer loaded activated carbon (AC) for the water flow quantification.*" P5 L7

11.) Page 5 Lines 34-35, Page 6 lines 1-6: Were the HPFMs stored dry? When placing the HPFMs in the streambed there was a first wash of the resin and AC, could it be estimated how much is this first contact with stream water influencing the final result? How much of the maximum potential adsorption/dilution (%) is lost in this first step?

Yes, they were stored dry. "*HPFMs were built, stored dry and…*" P9 L21
We moved this section to a later part where it makes more sense and is better understandable. A "Flush" should be avoided by the deployment procedure. We clarified this in the respective paragraph! "*The diameter of the steel tube for installation tightly fitted with the rubber washers at the top and bottom end of the HPFM, so that vertical water flow through tube and HPFM during installation was inhibited.* " P9 L26
In addition a control HPFM was built to assess the potential loss/nutrient accumulation during deployment and retrieval. "*One additional HPFM with alternating layers was used as a control HPFM, in order to assess potential tracer loss or nutrient contamination during storage, transport and deployment/retrieval.*" P10 L17ff

12.) Page 6 Line 14: The heading of this section is confusing since it is commonly placed at the end of the methods section; however this is a methodological manuscript. This section would be better merged with the correspondent method, Section 2.2.1 included right after the description of the AC, and section 2.2.2. merged with the description of the resins.

We rearranged the method section

13.) Page 7 Lines 4-6: Is "*JN*" time-averaged advective horizontal nutrient flux? Please indicate it together with the correspondent units.

completed "*time-averaged advective horizontal nutrient flux $J_N$ (mg m² d$^1$) can be calculated*" P8 L24

14.) Page 7 Line 10: The heading of section 2.3. is a bit confusing, does not reflect the aim of the section, to ease the reading, it might be better to swift this section to right after section 2.1.1.

As mentioned above, the method section was rearranged

15.) Page 7 Line 11: If as indicated experiments described in that paragraph were accomplished on triplicate; please present the data as means +/- standard error, or standard deviation.

std deviations or ranges were added in the text and tables

16.) Page 7 Lines 16-29: Please, provide more details on the experimental setup, for instance were the columns pump bottom-top or top-bottom direction, where the columns placed vertically or horizontally, for how long were the tests run. Please provide the brand of the pump.

We added information to this paragraph "… *placed in a vertical position and infiltrated with water collected from the study reach. … in order to ensure uniform infiltration at the surface of the column. Water was continuously pumped (peristaltic pump, ISMATEC® BVP Standard, ISM444) through the columns from top to bottom for 22 days at a speed of 20 mL $h^{-1}$, which also equals the expected Darcy velocity of $q_x = 4$ m $d^{-1}$. River water was supplied from a 22 L HDPE canister (Rotilabo® EPK0.1). SRP and $NO_3^-$ concentrations in this reservoir were revised daily. The draining water at the bottom outlet of the columns was sampled twice a day and analyzed for SRP and $NO_3^-$* P6 L21ff

17.) Page 7 Line 19: I wonder whether at the same nutrient flux into the HPFMs (high concentrations and low flow, or low concentrations at high flow) the differences in interstitial velocity (i.e. Darcy velocity) would influence the adsorption/dilution due to turbulent flow and finer diffusive boundary layer.

We do not think so and laboratory experiments I earlier studies, did not indicate that there are biases in this direction (see Cho et al. 2007). However, a validation on "field-behavior" should be conducted as soon as more measurements during different flows (especially low darcy flows) are available.

18.) Page 7 Lines 30-32: Please provide these results.

We provide the results in the supplement (APPENDIX A) and indicate that those supplements exist at the relevant point in the results section

19.) Page 8 Lines 1-2: The lower or higher concentrations of N and P respectively contained after the incubation were used to correct the obtained data in the field experiment?

The control HPFM was used to correct the HPFM result, because the control also accounts for other methodological errors. "*The results from the control HPFM also include uncertainties arising from sample storage, analytical processing and the background concentration of nutrients on the resin. Measurements of the other HPFMs were corrected by subtracting the transport, storage and deployment related tracer loss and nutrient accumulation detected in the control.*" P10 L20ff

20.) Page 8 Line 17: Does "stones" refer to Boulders or cobbles? Please specify. If possible, please provide information on the granulometry of the streambed.

Completed." *The sediments at the selected site are sandy with gravel and small cobbles. Sieving of sediment samples delivered the effective grain size $d_{10}$= 0.8 mm and a coefficient of uniformity $C_u$ = 3.13. The effective porosity $n_{ef}$ is 13 %. After Fetter (2001) the intrinsic permeability can be estimated to $K_i$ = 96 m² and the hydraulic conductivity to k = 81 m day$^{-1}$ Clay lenses are present in the deeper sediments below 35 cm.*" P9 L10ff

21.) Page 8 Line 33 Which is the reasoning for doing such combination of resin and AC? Would the results be more accurate in this way? If the aim is to test simultaneously both approaches, this arrangement does not seem appropriate since each layer is considered independent from the other one, and it is not clearly assumed that the streambed will have uniform nutrient concentrations or interstitial flows.

We tested two different approaches, both with inherent advantages and disadvantages. Depending on study site and research question the one or other might be preferable. We explain this in the discussion. There we also point out that the heterogeneity of the hyporheic zone has to be considered. P15 L3ff

22.) Page 9 Lines 5-7: Are the presented results corrected with this control?

Yes, see 19.)

23.) Page 9 Lines 19-20: it is not clear why the oxygen loggers had to be placed four weeks in advance for re-equilibration, while the HPFMs where placed without re-requilibration period. Would it be wise for future measurements to leave for example a perforated metal case in the streambed for certain period before placing the HPFMs? In this way, would the hyporheic zone be re-equilibrated after hammering the metal case in the streambed?

Different to the oxygen loggers or the MLS samplers the HPFM did not have an impermeable outer casing. Also the relation between installation time and measurement time is different. We explain this in the discussion

"*Unlike typical well screen deployments where PFMs (Annable et al., 2005; Verreydt et al., 2013) or SBPFM (Layton, 2015) have been inserted into a screened plastic or steel casing, our technique enabled the direct contact of the HPFMs with the surrounding river sediments. Thereby, the integration of the HPFM in the natural system is improved and the generation of artificial flow paths along the wall of the device is avoided. As a result, the disturbance created by the HPFM is low compared to other intrusive measurements of hyporheic flow like a piezometer or salt tracer injection. Additionally, the HPFMs include a measurement time that is long relative to the duration of the installation*"

P15 L32ff

24.) Page 9 Lines 21-37 and Page 10 Lines 1-3: The manuscript aims to compare the HPFMs with the pore water sampler (MLS), therefore this has to be clearly stated and well explained in the methods.

As mentioned above, the primary aim of the study was NOT to compare these two methods directly. However, we agree that the description of the MLS sampling was incomplete and added additional information to this paragraph (eg. extraction rate). For more detailed information please see the cited literature: Sänger and Zanke (2009)

"*Per sampler and depths 10 mL of pore-water was manually extracted by connecting a syringe to the open end of the Teflon tube and slowly sucking up water at a rate of 2 mL min$^{-1}$. The 4 extraction depths were sampled successively, always starting with the shallowest depths and continuing with ascendant depths. Manual pore-water samples were taken on the 4$^{th}$ and 11$^{th}$ of June 2015, both times between 1 pm and 4 pm local time.*" P11 L20

25.) Page 9 Line 23 and figure 2: Why is the MLS A located so far (>2m) from the rest of the measurement points?

We actually had more samplers installed in a transect. Unfortunately, the others were destroyed by vandalism so that we had only those two left for the experiment.

26.) Page 10 Lines 1-3: Are the N and P data from June presented in the manuscript? Or for June just information on SO3 and B are provided? And data from N and P just correspond to October?

We admit the information was confusing and clarified this paragraph "*As $NO_3^-$ and SRP concentrations in the pore water samples taken on June 4[th] and 11[th] 2015 were unexpected and inconsistent with results from the HPFMs, the sampling was repeated on the 8[th] of October. The aim of this repeated sampling was to investigate whether diurnal variations in subsurface $NO_3^-$-N and SRP concentrations could explain the discrepancies between MLS and HPFM results. In October, both MLS were sampled twice, the first time in the early morning before sunrise and again in the early afternoon (around 2 pm). Those samples were analyzed for $NO_3^-$, SRP and $SO_4^{2-}$. Due to technical issues, B could not be measured in October.*" P11 L31ff

27.) Page 10 Lines 4-13: The measurements presented here should have an appropriate heading as the MLS, oxygen profiles… or do these methods belong to the MLS?

We agree! "*Surface water chemistry*" P12L1

28.) Page 10 Line 7: Within the context of the manuscript it is also interesting to provide the detection limit.

The manufacturer supplies the detection limit of 0.03 mg L-1, which is noted in the article as "precision" for the Pro PS probe. We added a line on LOD definition, to avoid confusion on terms. *"Instrumental precisions refer to the limits of detection (LOD) as stated by the manufacturers" P5 L23*

Anyway, we are measuring at concentrations above 2 mg NO3 per liter, so we don't think that the detection limit of the sensor will be relevant in our study site.

29.) Page 10 Lines 10-13: Most of the parameters measured with the YSI probes are not provided in the results or tables. Please include them in a table, with mean and standard error or deviation for the incubation period.

We added table 4

30.) Page 10 Lines 12-13: Which is the relevance of Chlorophyll-a for the aim of the manuscript?

We agree that it does not make sense to mention parameters just because the probe can measure them. We deleted chlorophyll a from the article.

31.) Page 10 Lines 17-19: Indicate clearer, if correct, the abbreviations of all terms: "the proportion of surface water (QSW, m3 s-1) infiltrating…"

We clarified terms and completed units

32.) Page 10 Lines 17-19: Considering the interesting information about Darcy velocities in the hyporheic zone provided by the HPFM, it would be more accurate to calculate the proportion of infiltrated surface water from the cumulative QHZ for each layer, so the ratio will be ΔQHZ/QSW.

Since one of the advantages of the HPFM is that it can measure nutrients and Darcy velocities simultaneously and at different depths, the results will be more complete using an approach that includes that information.

*We agree that this would deliver additional information and is a further option for future uses of HPFM (which we will discuss in the discussion section). We only have very few measurements, so that calculations of this kind would be speculative. However, we liked your idea and included an example for depths wise calculation of uptake rates to the discussion: "Calculating $U_{NO3-HZ}$ in the same way for each single depth assessed with HPFM can deliver additional information about vertical gradients on nutrient processing rates and help to identify the most active depths in hyporheic zone. $U_{NO3-HZi}$ of a particular layer in the hyporheic zone can be derived by the differences in uptake rate between the regarded layer and the overlying layer. For instance the removal rates attributed to the different layers of HPFM L6 would be $U_{NO3-HZ15} = 567$ mg $NO_3^-$-N $m^{-2}$ $d^{-1}$ in the shallow layer (0 to 15 cm depths), $U_{NO3-HZ30} = 174$ mg $NO_3^-$-N $m^{-2}$ $d^{-1}$ in the layer from 15 to 30 cm depths and $U_{NO3-HZ45} = 256$ mg $NO_3^-$-N $m^{-2}$ $d^{-1}$ in the deepest layer from 30 to 45 cm depths. From this example one could conclude that the shallowest sediments are the most efficient ones in term of nitrate removal. While removal activity is first declining with depths it later increases again. This finding is consistent with the higher amplitudes of oxygen concentration in 45cm depths compared to 25 cm depths, also suggesting higher biotic activity at the deepest layer. Potential reasons for this pattern could be decreasing nitrate penetration with depths (lower uptake at the middle layer than the shallowest one) which is in the deepest parts counter balanced by increased residence time and stronger reducing conditions." P17 L37ff*

33.) Page 10 Line 18: I am not sure if the measured velocity in the HPFMs can be described as horizontal it could have also been diagonal. Of course according to the calculation it is horizontal but it confuses the reading especially when the results from the temperature show that there was a very strong vertical downwelling. Another term such as interstitial velocity may be more appropriate.

We agree, this term is misleading here and changed it to horizontal vector of the Darcy velocity. The angle of hyporheic flow was assessed as well. See section 3.2.1. Vertical Darcy velocity ($q_v$)

*With this, vertical flow $q_y$ was slightly lower than average horizontal flow $q_x$. Resulting from the relation between $q_y$ and $q_x$ the angle of hyporheic flow ($tan\alpha = \frac{q_y}{q_x}$) was 32° downwards" P13 L22ff*

34.) Page 10 Lines 20-21: Due to the lag between the water entering the hyporheic zone from the surface and the measurements in deeper layers, it is no easy to calculate the removal of any nutrient in the hyporheic zone. The N removal activity of the hyporheic zone, as calculated, seems to underestimate the capacity of the hyporheic zone. Considering the interesting information from each layer provided by the HPFMs, it would be interesting to take advantage of it and calculate the removal as something like what follows:
Since there is a quite strong vertical flow, we can assume that the concentration in one layer depends on the previous one. In this way, it can be calculated that the uptake at each layer results from the difference in fluxes (layery – layery +1). Of course, as indicated in the introduction, one has to take into account both flow and concentration, that is why it seems better to use the fluxes and not the concentrations. The combined uptake of all layers will provide the total uptake rate in the studied section of the hyporheic zone that can be then compared with the N flux from the surface water.
Additionally, it could be calculated the amount of N removed in the hyporheic zone to the flux of N in the stream to have larger scale information. This would answer the question of how much does the hyporheic zone removes from what is in the stream/ecosystem?

Please see point 32.)

35.) Page 10 Line 31: How could this influence the results? Please include some data.

Data are included in APPENDIX A which is indicated in this paragraph.
The impact of biofilm on the measurement is discussed in the discussion section.

36.) Page 11 Lines 33-34: See comment on page 5 lines 11-12, could this observation explain, at least partially, the measured darcy velocities (figure 3) in the deeper layers?

We do not think so. AC 4 and AC 2 do not look so different.
Clay will be definitely less permeable than the HPFM/mesh. However, you are right that a statement like this (if relevant) needs further discussion. Since we didn't see any effect of the clay lens we removed this statement from the article

37.) Page 12 Lines 4-7: The high variability between both measurements (A and B), the lack of

replicates and hence the lack of statistical tests, makes it difficult to draw such conclusions out of the presented data. The presented values represent very high spatial variability in the hyporheic zone, either due to heterogeneous flow or the presence of hotspot/moments during the day. A more cautious sentence should be used, and in the discussion refer to data from the literature.

We changed the formulation and added a sentence admitting, that the differences are high! "*In the repeated manual pore-water samples taken in October (**figure 6**) $NO_3^-$ concentrations were uniformly higher in the early morning than in the afternoon, whereas SRP behaved the other way round. This trend was consistent in both samplers even though the average concentration and distribution over depths differed between the samplers A and B.*" P13 L35ff

38.) Page 12 Lines 8-9: Are these values means or just punctual measurements, standard deviations should be then provided. If enough data are available, a simple statistical test should be applied, for instance, one-way analysis of variance (ANOVA).

Continuous readings from sensors. We added table 4, including std, min and max values which we think is more informative here than a one way ANOVA

39.) Page 12 Lines 13-14: The temperature profiles provided information on vertical downwelling from the surface water. However, the darcy velocity obtained from the HPFMs was named as horizontal, however it is not possible to know which the actual direction of the water was. To avoid confusion with the fact that the flow was strongly vertical it might be better to name the flux obtained in the HPFMs as interstitial velocity and assume the sediment was isotropic.

What the HPFMs really measure is the horizontal vector of the interstitial flow. Likewise in the temperature profiling we assess the vertical vector of the interstitial flow. We clarified this in the method section, explaining the vertical flux measurements: "*The vertical vector of hyporheic Darcy velocities $q_y$ were measured supplementary to the horizontal fluxes assessed with the HPFM in order to estimate the general direction of flow (upwards or downwards) and to calculate the angle of hyporheic flow.*" P10 L27ff

40.) Page 12 Lines 19-23: Considering the high variability shown with two HPFMs in the manuscript, at least three per parameter should be placed in the streambed (flow or nutrients). Even in channelized rivers, small scale variability and heterogeneous residence time distribution occurs (see for example data from vertical water flux Mendoza-Lera, C. and M. Mutz (2013). "Microbial activity and sediment disturbance modulate the vertical water flux in sandy sediments."

Freshwater Science 32(1): 26-35.). Additionally, even if low variability is assumed, such approach would be statistically more consistent.

We agree and admit the lack of sufficient samples for quantitative statements at several points while underlining the need for a higher density of measurements "*Even in those systems, small scale variability in stream bed and sediment characteristics can cause spatially heterogeneous flow distributions (Lewandowski et al., 2011; Mendoza-Lera and Mutz, 2013). The second approach with alternating nutrient sorbents and water flux measuring segments is therefore preferable in most other cases as long as a high resolution over the vertical profile is not required. In general, several HPFMs should be grouped together in order to obtain representative results.*" P15 L8ff

41.) Page 12 Lines 30-31: Data on substantial biofilm growth are not provided, please include.

See APPENDIX A

42.) Page 12 Lines 31-34: Not only biofouling could influence the results, what about uptake/release of nutrients by the biofilm? Right after placing the device in the streambed in was a sterile substrate, which informed about the nutrient concentration in the water flowing through it, and therefore about the surrounding conditions in the hyporheic zone. However, after certain time the HPFMs become an actual physical substrate where the microbial community developed, and therefore the HPFMs became part of the hyporheic zone. Therefore, the information provided by the HPFMs after the incubation also refers to the community inhabiting it.

That's correct! And an important issue which we incorporated in the discussion. As mentioned above, the biofilm growth on the resin granules remains the mayor limitation to the method which we did not outline that well before but make clear in the edited version.
"*We observed substantial biofilm growth on the resin in the laboratory and on the top 2 cm of the field-deployed HPFM R2. The results of the column experiments suggest that biofilm growth on the resin porous media did not affect its loading capacity and that biofilm growth only started after the loading capacity of the tracer was exhausted. R2 detected higher $NO_3^-$ fluxes in the top layer than the other HPFM. This could be due to contamination of the top layer of this HPFM with surface water (if the HPFM was not introduced sufficiently deep into the sediments), this would further imply that this layer was exposed to much higher water and nutrient infiltration, so that the loading capacity was exhausted before the end of the experiment, thus allowing biofilm accumulation. At the current state it is unclear, to what extent the biofilm bound nutrients can be extracted by the procedure used here*" P15 L18ff

43.) Page 13 Lines 4-6: Please provide example of the intrusive measurements of hyporheic flow. As occurs when placing piezometers of smaller diameter than the HPFMs+metal case disturbance is created and likely after removing the metal casing fine sediment was sucked into the HPFMs as happens when placing piezometers. When the HPFMs were removed from the streambed and the measurements perform, was there evidence of fine sediment intrusion? Was there any evidence of clogging in the mesh?

We are more precise and carful on this comparison now.
We did not observe sediment intrusion, neither clogging of the mesh.(see also 19.) But we agree,that this is a point which should be considered and which we mention in the discussion. We also clarify that potential convergence or divergence into/around the device is accounted for in the equation and outline the limitations for this correction. Also see introduction "*Corrections for convergence and divergence of flowlines into or around the flux meter have been established in earlier studies (Klammler et al., 2004). However, accounting for an impermeable outer casing of a flux meter is much more complicated and requires additional factors which have to be determined experimentally for each specific application (Klammler et al. 2004, Annable et al. 2005, Hatfield et al. 2004). For hyporheic studies we therefore intended to deploy the passive flux meter in a way that allows direct contact with the surrounding sediments and minimal manipulation of the natural flow pattern*" P4 L27
and method section 2.5.2

44.) Page 13 Lines 11-17: It would be interesting to know which is the time scale detected by the method. If oscillation occurs within 12 hours or less, could the method be further adapted? For example placing the HPFMs for few hours?

We included a discussion on upper and lower limits "*The minimum and maximum deployment time will depend on the Darcy velocity and nutrient concentrations at a study site. As the values derived from the control incorporate all the processing steps of HPFMs and samples, they can be regarded as the method detection limit MDL (Greenberg et al.,1992). The MDL defines the lower limit for the use of HPFMs in in cases were nutrient fluxes are very low and deployment time cannot be extended. We recommend that a control HPFM be incorporated in each field application of HPFMs in order to determine the specific MDL. The upper limit is given by the loading capacity of the resin or complete displacement of all resident alcohol tracers.*" P14 L31ff

45.) Page 13 Lines 30-32: I am not sure whether such assertions can be done in the light of the limited replication and statistical tests. Since the difference in P concentration has not been proofed (no statistics) it is no possible to link the dynamics of phosphorous with oxygen. However, it could be interesting to take advantage of the oxygen profiles and determine if the dynamics observed in N and P in the HPFM and/or MLS correlate with the mean oxygen concentration.

*We agree that this was rather speculative and reduced this statement to "The redox conditions in the subsurface also regulate the mobilization/demobilization of phosphate (Smith et al., 2011). The repeated manual sampling of pore-water from MLSs in October showed diurnal variations of SRP and $NO_3^-$ in the subsurface of the testing reach, supporting the hypothesis that diurnal cycles in benthic metabolism caused temporal variations in hyporheic SRP and $NO_3^-$ concentrations at our study site." P16 L22*
*We tried to assess the correlation between oxygen concentration and nutrient uptake by a bi-dial sampling of MLS, the results are discussed as indicated above*

46.) Page 13 Lines 36-37: The HPFMs provide information of a flowpath, the length, velocity and residence time of the water before reaching the HPFMs is not known (further tracer tests could be implemented in combination with the HPFMs). Then it is difficult to determine whether there are hot spots for denitrification or hot moments, or both (see Abbott, B. W., et al. (2016). "Using multi-tracer inference to move beyond single-catchment ecohydrology." Earth-Science Reviews). Additionally, downwelling into the hyporheic zone does not occur as a front but rather as semicircular flowpaths, see for instance:
Thibodeaux, L. J. and J. D. Boyle (1987). "Bedform-Generated Convective-Transport in Bottom Sediment." Nature 325(6102): 341-343.
Salehin, M., et al. (2004). "Hyporheic exchange with heterogeneous streambeds: laboratory experiments and modeling." Water Resource Research 40: W11504.
Rehg, K. J., et al. (2005). "Effects of suspended sediment characteristics and bed sediment transport on streambed clogging." Hydrological Processes 19(2): 413-427.\

*We are more carful on our formulation now*
*"We found continuously degreasing $NO_3^-$ concentrations with depths, suggesting that this entire area (and potentially deeper) of the subsurface contained active sites for denitrification" P17 L15*

*We are aware that (vertical) water flow will be heterogeneous and much more complex than a singular flow direction, however we think that our conclusion on the extension of the hyporheic*

zone is still correct. We clarify that this is an assumption and suggest additional tracer tests to accomplish HPFM measurements. "*Conducting collateral tracer tests, as suggested for example by Abbott et al. (2016), could deliver further evidence and characterize distinct flow paths. Nevertheless, since vertical water movement was overall downward and the lowest concentrations of NO3- were observed in the deepest segments of the HPFM, it is very likely that the hyporheic zone at our study site extends deeper than the 50 cm evaluated*" P17 L19ff

47.) Page 14 Line 10: I suggest including in this section an summary in the form of a table of the factors that should be taken into account for applying the HPFMs in different contexts, for example pH should be taken into account in an acidic stream in a mining area, or permeability is to be taken into account to approximate the permeability of the studied reach (for instance the following method could be used to define the appropriate permeability in the HPFMs Datry, T., et al. (2014). "Estimation of sediment hydraulic conductivity in river reaches and its potential use to evaluate streambed clogging." River Research and Applications 31(7): 880–891).

The overall aim was to develop HPFMs which can be applied in a wide range of systems. Changing the sorbent would require to repeat a lot of analytical work on the new sorbent in order to identify the sorptive characteristics, retardation factors etc.  The permeability of the HPFM (relative to the surrounding environment) is incorporated in the correction factor alpha. We recognize that this issue rose questions in the reader. We provide more information about this factor in the method section and added a line to the discussion. For more details please see the cited articles (Hatfield et al 2004, Annable et al 2005)
"*The correction for convergence of flowlines into the device or divergence around it is relatively simple and already incorporated in the equation for the flux calculation. We believe that it is applicable for a wide range of field conditions. However, for very coarse sediments, a protection of the HPFM with a screened plastic or steel casing might still be preferentia*l" P16 L3ff

48.) Page 14 Lines 2-5: I agree, the hyporheic zone is probably deeper than the scale of the HPFMs, however it is as well very heterogeneous in residence time distribution and therefore increasing the number of measurements and/or the scale will be a very interesting. This, together with the high variability of the provided data, indicates that future applications of HPFMs should have enough replicates.

Yes, we have to admit that!

*"Considering the high spatial heterogeneity of the hyporheic zone, a higher number of HPFM would be needed to derive reliable and statistically supportable rates of hyporheic nutrient dynamics. The following example aims to display further possibilities of interpreting HPFM measurements. At our study site,…"* P17 L24

49.) Page 15 Line 20: When more than one author it is uncommon to acknowledge in first person.

was changed

50.) Table 2: Please include either ranges or means +/- standard deviation.

completed

51.) Table 3: Please include either ranges or means +/- standard deviation.

completed where possible

52.) Figure 1: Please indicate to which approach corresponds the picture resin or AC, or alternating segments.

Figure capture improved

53.) Figure 2: For a non-german reader this map won't be very informative, especially because the aim of the manuscript is not to study that stream. I encourage presenting other information instead such as the temperature profiles.

We removed this map from the figure. However, we don't think that a grave of the temperature profiles is very informative. We reconsidered which results we would like to present as graphs and decided not to add another graph. We did supplement figure 1 in order to make the functioning principle of HPFM easily understandable.

54.) Figure 3: It might be more appropriate to express the fluxes per volume of sediment, in this way confusion with the denitrification rate units would be avoided. Additionally, in table 2, mass fluxes for nutrients are expressed with other units.

units corrected

55.) Figure 5: I am not familiar with the symbol for diameter to indicate mean concentrations, and it is a bit confusing. I suggest to simply adding surface water.

Considering the variability of the values measured in the hyporheic zone, it would be interesting to include the range of concentrations, or mean +/- standard error or deviation, of the surface water during the deployment time, as understood from the method the nutrient concentrations were measured every 15 mins.

Ø was replaced by "average", min and max concentrations were added.

**Technical corrections**

56.) Page 2 Line 18: Correct phosphate, per phosphorous, or P per PO4-

57.) Page 3 Line 20: Correct technics, per techniques.

58.) Page 4 Line 34: Space missing in "…Pin…"

59.) Page 7 Line 35: Correct KCL per KCl

60.) Page 8 Line 13: Correct figure 2 instead of figure 3

61.) Page 10 Line 2: Correct sun rise, per sunrise

62.) Page 10 Line 20: Correct NO3-, per NO3--N

63.) Page 12 Line 4: Correct SRP, per SRP-P

64.) Page 12 Line 14: For which sentence stands the citation (Layton, 2015)?

65.) Page 14 Line 20: Correct hot spot, per

**J. Lewandowski (Referee)**

lewe@igb-berlin.de

Vanessa Kunz and coworker present in the manuscript "Quantifying nutrient fluxes in Hyporheic Zones with a new Passive Flux Meter (HPFM)" a novel technique to measure horizontal water fluxes and nutrient fluxes in hyporheic zones. Without doubt this is an exciting technique to answer unsolved questions about transport and turnover in hyporheic zones. Up to now the lack of adequate techniques hindered in-depth investigations of transport and turnover in this important transition zone.

Dear Jörg Lewandowski,

thank you very much for your comments and suggestions! Your remarks contributed productively to the improvement of our article!
Please see below our responses to your comments.

Major comments

My major concern about this method is its impact on subsurface flow paths and flow velocities.
As a consequence the calculated loads might be misleading. Since the device will be placed in sediments with different hydraulic conductivities, its hydraulic conductivity will sometimes be larger than that of the rest of the sediment and sometimes smaller. In cases where the hydraulic conductivity is lower than that of the surrounding sediment most flow paths will bend around the device instead of passing it. As a consequence there will be much less uptake of nutrients and smaller flow rates. In the opposite case flow will be "sucked" into the device. Nutrient uptake and flow rates will be overestimated. This question could have been addressed in a lab experiment with a box filled with different sediment types with known horizontal flow and a HPFM placed in the center of the box. Alternatively, modelling would also be a method to address this problem. Even without additional investigations it is necessary to carefully discuss this shortcoming and even mention it in the abstract.

Potential convergence or divergence into/around the device is accounted for in the equation (2) for $J_N$ with the correction factor $\alpha$. We admit that this was not adequately presented in the original manuscript. We completed the method section on nutrient fluxes (Chapter 2.5.2.), explaining this correction factor and referring to literature for more detail:
P8 L27 ff "…*and $\alpha$ (-) is a factor ranging from 0 to 2 that characterizes the convergence ($\alpha > 1$) or divergence ($\alpha < 1$) of flow around the HPFM. If, like in the case presented here, the hydraulic conductivity*

*of the HPFM sorbent (resin or AC) is much higher than of the surrounding and the HPFM is in direct contact with the sediments (i.e. in absence of an impermeable outer casing or well wall), $\alpha$ can be estimated after Strack and Haitjema (1981)*

$$\alpha = \left( \frac{2}{1+\frac{1}{K_D}} \right) \qquad\qquad (3)$$

*where $K_D = k_D\, k_0^{-1}$ is the dimensionless ratio of the uniform hydraulic conductivity of the HPFM sorptive matrix $k_D$ ($L\ T^{-1}$) to the uniform local hydraulic conductivity of the surrounding sediment $k_0$ ($L\ T^{-1}$). For more details on the correction factor $\alpha$ and applications where a solid casing is required or the permeability of the surrounding sediments is higher than of the device see Klammler et al. (2004) and Hatfield et al. (2004)"*

We also discuss potential limitations for this correction, P16L3ff: "*The correction for convergence of flowlines into the device or divergence around it is relatively simple and already incorporated in the equation for the flux calculation. We believe that it is applicable for a wide range of field conditions. However, for very coarse sediments, a protection of the HPFM with a well screen might still be preferred.*"

Background information on the correction is additionally presented in the revised introduction. P4 L27ff "*Corrections for convergence and divergence of flowlines into or around the flux meter have been established in earlier studies (Klammler et al., 2004). However, accounting for an impermeable outer casing of a flux meter is much more complicated and requires additional factors which have to be determined experimentally for each specific application (Klammler et al. 2004, Annable et al. 2005, Hatfield et al. 2004). For hyporheic studies we therefore intended to deploy the passive flux meter in a way that allows direct contact with the surrounding sediments and minimal manipulation of the natural flow pattern*"

Minor comments

1.) In general the paper is very well written and I could only identify a few typing and grammatical errors.

We corrected mistakes marked by you and other reviewers and double checked the revised version

2.) The only section of less quality is the abstract. I had the feeling that this was written in a hurry after finalizing the rest of the manuscript. However, as most central part of the manuscript it deserves more care to assure the high quality of the rest of the manuscript.

We edited the abstract concentrating more on the method development, also mentioning limitations (e.g. biofouling) of the method.

3.) The introduction is well written but relatively long. You might consider to slightly shorten it.

We condensed the introduction, focusing on the methodological aspect of this study. We therefore also added information where required (eg. on conversion/divergence of flow lines around the device)

4.) The material and methods section is also relatively long and sometimes a bit confusing. Consider to improve its structural elements.

As we focus mainly on the method development, it is inevitable that the method section is detailed and as a result long compared to the other sections. We recognized that the structure of this section was confusing to several reviewers and reorganized it.

5.) The results section is short. The discussion is very well written and of optimum length. The same applies to the conclusions section.

The aim of this study was not to characterize the processes at our study site, that's why we think that the presented results are sufficient. We also considered moving parts of the discussion (eg. the uptake calculations) or methods (the results from the background nutrient extraction) to the results, but finally decided that they were more appropriate at their current location

6.) The left part of Fig. 2 can be removed. I do not see any need for this figure and it is so small that it is impossible to see anything here. I recommend refering to another paper where such a map has been included instead of this one here.

We removed this part of the figure and referred to Kamjunke et al., 2013, where the study stream is explained in more detaile.

7.) P2L24ff: "In stagnant waters, such as lakes, the transport of dissolved nutrients to the sediments is dominantly controlled by diffusion. Therefore, surface water concentrations of nutrients are a good predictor for uptake processes and potential limitations (Dillon and Rigler, 1974;Jones and Bachmann, 1976)." Both sentences are completely wrong. Diffusion is only a relevant transport process over very short distances but not in a water body or as transport process from a lake to its sediment. Diffusion is a relevant process in the diffusive boundary layer above the sediment

surface. In the water body there are many active transport processes such as wind- and temperature-induced transport. You can also discuss this with the people at UFZ Magdeburg involved in lake physics. Transport of nutrients to the sediment occurs mainly in particulate (and not in dissolved) form. In lake sediments many different advective transport processes occur in addition to diffusion, for example groundwater discharge, wave- or seiches-induced pore water transport in the sediment and bioturbation. The latter is especially relevant in shallow lakes. For example, for Lake Müggelsee in Berlin it is well-known that chironomids pump the entire water body through the sediment once a week. Referring to the second of the above cited sentences: Nutrient concentrations in surface waters of lakes are mainly controlled by processes in the water column. For example SRP concentrations are controlled by the very efficient uptake of SRP by plankton during the growing season. That is the reason why SRP concentrations even in eutrophic lakes are usually low during the summer.

We agree you are right and deleted this paragraph, refocusing the introduction in general

8.) In the introduction I would expect a paragraph about hyporheic flow and methods to measure hyporheic flow. The HPFM aims on measuring water and nutrient fluxes but the introduction solely focusses on nutrient fluxes. Please add something about determination of hyporheic flow. For example the heat pulse sensor of Lisa Angermann could be mentioned here but also other methods. I know that the HPS did not work at your site but nevertheless you can mention that there is a device that can be used in finer sediments.

We completed the introduction P3 L23ff: "*Exchange rates are traditionally assessed via hydraulic head differences or tracer injections (USEP 2013, Fleckenstein et al. 2010). High resolution vertical temperature profiles have efficiently been used to derive vertical Darcy velocity (qy) (m d$^{-1}$) in the streambed. This method is based on time series measurements of temperature in the stream and in the sediments at several depths. Based on a numerical model, vertical flow velocities can then be calculated from the measured attenuation and phase shift of the diurnal temperature signal which, at depth, varies with the vertical hyporheic flux (Keery et al. 2007, Schmidt et al. 2014). While measurements of vertical Darcy velocities are a valuable asset and have been used as supplement in this study, horizontal fluxes are also needed in order to assess hyporheic transport and residence time (Binley et al. 2013, Munz et al. 2016). Active heat-pulse tracing enables highly resolved in situ measurements of direction and velocity of hyporheic flow (Lewandowski et al., 2011; Angermann et al., 2012). These methods are valuable in shallow sediments (max.15-20cm) and rivers with fine sediments, but may not be implementable in streams with coarser sediments.*"

9.) In general I think you consider the sediment only as a sink and not of a source of nutrients. For example in line 11 on page 1 you write "nutrient removal" although the hz is sometimes a source. Besides a temporary storage (e. g. uptake and later release of nutrients) the transport of particulate organic matter to the sediment surface should also be considered. Once this organic material is buried it can release nutrients as a consequence of mineralization processes. Keep this in mind throughout your manuscript.

We changed formulations to "*nutrient processing*" were adequate. While of undeniable importance, particulate organic matter transport was not in the scope of our study (because only solutes are assessed with our methods). With the aim of condensing the introduction to the points really relevant for HPFM measurements, which is hyporheic solute fluxes, we decided to exclude these points from our article.

10.) The sediment description on page 8 is quite poor. It would be great to know a little bit more about the grain distribution or hydraulic conductivity of the sediment.

We completed the description of the sediment characteristics P9 L10ff "*The sediments at the selected site are sandy with gravel and small cobbles. Sieving of sediment samples delivered the effective grain size $d_{10}$= 0.8 mm and a coefficient of uniformity $C_u$ = 3.13. The effective porosity $n_{ef}$ is 13 %. After Fetter (2001) the intrinsic permeability can be estimated to $K_i$ = 96 m² and the hydraulic conductivity to k = 81 m day$^{-1}$ Clay lenses are present in the deeper sediments below 35 cm.*"

11.) P12L19ff I do not see the usefulness of this device. I think there is always much more small scale variability in hyporheic zones than assumed. Even in channels. I would recommend placing two alternating HPFMs in the sediment instead and place them with a small depth variation. In that case you will end up with the same spatial resolution but you can assure that flow and nutrient fluxes match to each other. Also, in the entire paper I have not understood the motivation for the separated HPFMs.

We evaluated two different approaches to construct HPFMs in a way that separates resin and AC and thereby prevents the nutrient background on the AC from biasing the results. We added an explanation to the revised article
P15 L3 ff "*The high nutrient background on the AC required the separation of resin and AC in the HPFMs. We tested two different HPFM designs in this study, of which each inherits designated characteristics being more or less beneficial for different specifications:*"

*We believe that both approaches have advantages and disadvantages. "The first approach, pairs of two HPFMs where one is used to assess the water flux and the second to capture nutrients is preferable if a high resolution depth profile is needed (a heterogeneous horizontal flux in the vertical direction). Since this approach assumes that local horizontal heterogeneity is negligible in the range of 20-30 cm, we recommend this type for use in uniform systems such as channelized river reaches."*

However we agree that the local heterogeneity of stream beds/hyporheic flow should be considered and mentioned at this point: P 15 L3ff "*Even in those systems, small scale variability in stream bed and sediment characteristics can cause spatially heterogeneous flow distributions (Lewandowski et al., 2011; Mendoza-Lera and Mutz, 2013).*"

Summarizing, I recommend accepting the manuscript after minor revision.

Jörg Lewandowski, IGB Berlin
Please also note the supplement to this comment:
http://www.biogeosciences-discuss.net/bg-2016-334/bg-2016-334-RC3-supplement.pdf
**J. Rozemeijer (Referee)**

joachim.rozemeijer@deltares.nl

Review of Kunz et al, Quantifying nutrient fluxes in Hyporheic Zones with a new Passive
Flux Meter (HPFM).

The paper presents a useful monitoring tool to increase our understanding of nutrient cycling in the
hyporheic zone of streams. The paper is very well organized and written. I only have some minor
comments that could be considered by the authors.

Dear Mr. Rozemeijer,

thank you for your comments! We appreciate that you liked our article. However, for most other reviewers,
the structure of the article was confusing, that's why we rearranged several parts of it. Especially the
methods section.

We improved our article, also taking in account your suggestions.
Please see below our comments to your specific remarks.

P5 l6: From figure 1 the idea of tracer release and nutrient absorption is not immediately clear. Also, the
difference between the sections filled with tracer and with absorber is not clear in the graphic (same color)

We improved and supplemented Figure 1.

P6 l32 and P7: "if not indicated otherwise" Remove?

This phrase was removed

P9l8: At this point in the paper it is not clear why these additional measurements are also done. Consider
adding a short introduction.

We added a sentence or two to each of the sub-paragraphs briefly explaining the goal of each
measurement:

P10 L27ff *"The vertical vector of hyporheic Darcy velocities $q_y$ were measured supplementary to the horizontal fluxes assessed with the HPFM in order to estimate the general direction of flow (upwards or downwards) and to calculate the angle of hyporheic flow."* …

P11 L6ff *"We monitored the subsurface oxygen concentration as a primary indication on the redox status of the hyporheic zone in order to evaluate the potential for $NO_3$ reduction and $PO_4$ mobilization."* …

P11 L13f *"Pore-water nutrient concentrations were measured to substantiate the HPFM results."* …

P12 L2f *"Surface water concentrations of SRP and $NO_3^-$-N were monitored in order to compare surface and subsurface water chemistry."* …

P10l28: missing r in break

was corrected

P11l7: Here you mention declining concentrations with depth, but figure 3 shows fluxes. Adding concentrations to figure 3 would also be informative

We did measure flux ($J_N$) not concentration, so this was just wrong and is corrected now to $J_N$. Concentrations are not directly measured with the HPFM, however an estimate for the average concentration can be derived by dividing nutrient flux by Darcy velocity. These estimates are illustrated in Figure 5 (for comparison with MLS)

Also see P13 L31 *"In order to facilitate direct comparison, nutrient fluxes as measured in the HPFMs were converted to flux average concentrations which is the quotient of $J_N$ and the respective $q_x$"*

P11l13: For this conclusion (52% removal), you need to know that the vertical flux is downward and that groundwater has no impact on the concentration levels. However, the vertical flow is given after this conclusion. Re-order?

We moved the estimates on turnover and removal rates to a separate paragraph at the end of this chapter (see 3.2.2.)

P12l6: Higher should be lower?

Yes, you are right! We apologize!

P12l10: Discussion: You may consider to add a paragraph about the applicability of the HPFM. Can it also be applied to quantify vertical nutrient fluxes in lakes and other non-flowing surface waters? Is it applicable in case of a coarse grained hyporheic zone (stones, gravel)?

In general a different design is needed to assess vertical fluxes. Layton et al (2015) assessed vertical contaminant fluxes in river beds with PFMs. We mention this study in our revised article. P14 L26f "*An earlier study on passive flux meter (SBPFM) in river beds (Layton, 2015) only assessed vertical flow of contaminants and is therefore not comparable to the application presented here.* "

Also we added a line on the problems which might arise from coarse sediments

P16 L1ff "*While the installation of mini-drive points or heat pulse sensors in sediments coarser than sand is difficult or even impossible and also proved unfeasible at our field site, installation of the HPFM with the presented technique was successful. The correction for convergence of flowlines into the device or divergence around it is relatively simple and already incorporated in the equation for the flux calculation. We believe that it is applicable for a wide range of field conditions. However, for very coarse sediments, a protection of the HPFM with a well screen might still be preferred*"

P12l10: Discussion: The difference in concentrations measured in the MLS are quite different from the HPFM (figure 5). Is this only due to the diurnal variations? Other explanations? How do we know which method is the best one?

We added a paragraph, discussing the discrepancies between the two measurements (HPFM and MLS). In general, both measure different things (flux/concentration), so it will depend on the specific research question which is "the best" one.

P16 L10ff "*In June, we found discrepancies between the average concentrations measured in the HPFM and the concentration found using the MLS. From our measurements it is not possible to proof that the HPFM results are correct and the MLS results biased. However, the HPFMs showed the expected decline in JN, whereas in the MLS pore water concentrations were similar at all depths assessed. This can be related to two reasons: First, we sampled surface water which bypassed along the wall of the MLS in June but not in October. Second, we sampled the MLS at a time point, when the hyporheic zone was inactive in respect to nutrient processing. Considering the high diurnal amplitudes in hyporheic oxygen, we assumed that the discrepancy between HPFM and MLS arose from oscillations in hyporheic nutrient concentrations similar to the oxygen pattern.*"

P12l14: remove second(Layton , 2015)

was corrected

P12l18 two points at the end

was corrected

P14l15: Is this really permanent removal for PO4? Or can it later be released from its absorption sites?

was corrected to a *removal (uptake or adsorption)  rate for SRP*

**R. González-Pinzón (Referee)**

gonzaric@unm.edu

Please find my general and specific comments within the attached pdf.

I think this work has enormous potential to open an unexplored window of observation, but the manuscript has to be reorganized to convey a clear take-home message. I think that the main work to be done in the revision process is to define the actual scope of the manuscript, i.e., is this a methods paper with strong focus on the technological development? or is this a data-driven paper which presents novel, previously unattainable measurements (with a new technology) that revolutionize our understanding of biogeochemical processing?

Dear Mr. González-Pinzón,

thank you very much for your comments and corrections! Following the suggestion from you and other reviewers, we reorganized the structure of our article. As also noted by reviewers, the amount of data collected in our study (density of measurements) is not sufficient to make robust (e.g. statistically supportable) statements about the biogeochemical processes at our study site. We therefore decided to clearly focus our article on the evaluation of the methodology. The field study was mainly used to demonstrate the applicability of HPFM and to identify benefits and potential limits of the method. Besides that, we used the results to give an example how HPFM derived data can be interpreted. The revised manuscript clearly tracks that story.

We accounted for corrections of orthography or grammar where indicated by you or by other reviewers.

Please find our responses to your specific comments in the separate commented pdf file.

[revised manuscript text omitted]

---

## Referee Report (RR1)

[referee-annotated manuscript omitted]

---

## Author Response (AR2)

Kunz et al. "Quantifying nutrient fluxes in Hyporheic Zones with a new Passive Flux Meter (HPFM)" 2nd Revision

Response to Report No 1, Anonymous reviewer

The paper has been significantly improved, however I believe this resubmission still has significant issues and recommend some restructuring of the paper to highlight its goal and main contributions.

-The definition of the process measured has to clearly defined, in some parts of the text it is named as nutrient removal and others a denitrification.

**We double checked the usage of denitrification in the article and replaced it with "nitrate removal/uptake" where appropriate**

- There is a lack of discussion/introduction of the method used in groundwater. Since as indicated in the manuscript the HPFM is an adaptation from passive methods it would be informative to explain which aspects of the groundwater passive meter are to be adapted to be able to use them in the hyporheic zone. At first sight, there should be no problem to apply groundwater methods to the hyporheic zone, and thus needs to be clearly indicated in the manuscript.

We supplemented the last paragraph of the introduction with more information about the standard groundwater PFM. See P4 L6-35. We thereby focus on the main differences between the PFM and the HPFM: PFMs have not been used for nutrients under field conditions (only laboratory tests and only for PO4) and deployment under surface water requires a modified technique compared to deployment into standard land based well.

The introduction has improved but the message and goals are still unclear; they should be clearly stated to make the delivery stronger. I suggest re-organizing the introduction with to follow an structure similar to the following one, in order to get the reader to the final goal of the work:
Significance of the hyporheic zone. Include the significance of denitritication right here, so the all the examples can be easily understood from this perspective, there is lack of knowledge on any flux,

nitrogen is just an example easily used to test the HPFM.

2. Limitations imposed in the hyporheic zone to measure fluxes and rates: Two aspects have to be measured and combined in order to measure fluxes: flow and concentration.

3: Limitations of the commonly used methods to asses both components of the fluxes (including methods such as gels or peeper methods to determine pore water concentrations, which are like the primitives HPFM).

4. From a groundwater perspective, passive meter are being used to assess fluxes.

6. What is needed to apply those ground water methods to the hyporheic zone.

7. Goal of the manuscript.

We reorganized the introduction considering your suggestions.

The two primary limitations of existing methods are clearly stated now and a short note on peepers and gels is included. We do not discuss these methods more in detail here, because it was also suggested to condense the introduction.

Taking nitrogen only as an example is not what we intended. Nitrate is a reactive substance, subjected to several turnover processes and therefore assessing its dynamics may be much more challenging than measuring transport rates of an inert solute, e.g. the high variability of nitrate concentrations in surface and pore water are one reason, why snap shot sampling of nitrate concentrations are particularly problematic. Certainly, most techniques can be transferred between different solutes, but we also stress now that besides transforming a method designed for groundwater wells to the hyporheic zone conditions, adaptations were necessary to use a method for nutrients/nitrate.

- Some estimations of the applicability of the HPFMs would complete the interesting results and help future research, and would strengthen the message.

In short this is already in the discussion/ conclusion. See P19 L13-19. We had developed more ideas in an earlier version, but were then criticized that they were too speculative.

Specific comments

Abstract

P1, L28: Although the HPFMs can be certainly used to assess denitrification, the setup presented can only provide information on N-NO3 uptake, this uptake could result in denitrification (reduction of NO3 to N2) or in biomass production, use of N to build proteins, DNA... however this cannot be estimated from NO3 changes in concentration without additional N2 measurements or using isotopes like 15N.

Changed to "hyporheic nutrient dynamics, specifically nitrate uptake rates." P1, L33

P1, L31-33: Move the sentence to line 29 to have a more coherent abstract.

**Moved as suggested**

**Methods and Results**

To ease the reading, organize the methods and results section in a similar way, Laboratory tests and field tests. As well, use the same headings for the subsections that are present in both sections, for example subsection 3.2.2. should be named as 2.6.5, the first one is the direct results of the latter one.

**Headings revised**

P5, L20: If all the values are provided in those units, it would be more adequate and simple to use the terminology NO3--N or SRP –P so the reader does not need to remember that information. It is not indicated whether the PO4 values are expressed as PO4-P or PO4. Additionally, SRP is "only" P there is no need to indicate it, by contrast with NO3 measurements that provide the mass of the molecule NO3.

**The sentence was removed and NO3/PO4 were changed to NO3--N or SRP –P were appropriate**

P5, L21: "The experiments described in this chapter were accomplished on triplicate samples" It would be more informative to indicate the number of replicated when explaining the design of the experiment. For example in P2, L30, "... analyzing NO3- and PO4- from each resin (n = 3)".

We agree that adding (n=3) where relevant improves the information and deleted this introductory

**sentence**

P8, L1: Does this section refer to the field installation described in more detail on section 2.6? If yes, for the logic reading of the manuscript section 2.5 should be placed after section 2.6.

**Yes, we understand that this might be confusing and have moved it to a later section**

P11, L33-37: It is unclear the reason to assess the diurnal variation of with MLS within the scope of the manuscript. It is already indicated in the introduction that such sampling methods are highly influenced by spatio-temporal variability. Since the HPFMs were not used at that small temporal scale, it is needed to clarify the reasons behind those results. For example those results could support that the comparison between MLS and HPFMs result in higher values provided by MLS compared to HPFMs, since the latter one integrate those temporal oscillations. It would be of help to include a short sentence in the methods indicating that to assess the short term variability in nutrient fluxes...

We added: "We assumed that the HPFM measurements integrated temporal oscillations, while MLS samples represented the specific concentrations around noon. In order to test this hypothesis, both MLS were sampled twice, the first time in the early morning before sunrise and again in the early afternoon (around 2 pm) during the sampling in October." P11; L29

P12, L13: I suggest rewriting the title of the section to something like: Estimation of nitrate turnover with HPFMs

**Was changed to "Estimates of nitrate turnover rates based on HPFM measurements"**

P14, L9-10: As written, the sentence belongs to the discussion section, it might be better to merge it directly with the previous sentence and write, indicating low groundwater influence. If the significance of the measurements has to be clearly indicated, it might be better explained in the methods.

Was changed to "Likewise boron concentrations of 50 to 60µg L-1 were consistence with the

concentrations in the surface water, indicating only minor groundwater influence. Also in October  $SO_4^{2-}$  concentrations in the pore water samples were in the range of surface water concentrations, slightly declining with depth" P14,L5-8

**Discussion**

P14, L25-27: The reference to the study performed by Layton (2015) is a bit rough and the following sentences do not clarify the sentence. That Layton applied a similar method to assess contaminants does not mean that study performed is of less value. Instead, it would be better to explain how does the present work complements Layton work and which are the actual differences with Layton's work, for instance, did he used resins or AC? Did Layton also measured darcy velocities? In which type of hyporheic zone did Layton work, more permeable, less? A more detailed comparison will provide more information and will strengthen the message of the findings of the manuscript.

Layton developed a PFM for measuring seepage (vertical flux) of both water and contaminants in sediments. His work was conducted in controlled chambers in the laboratory. We added information on this reference in the introduction, at first mention. See P4 L17-20

P14, L19-21: The following sentence is not clear from the results why "... biofilm growth only started after the loading capacity of the tracer was exhausted". Biofilm growth starts as soon as the substrate is placed in the stream. Please, indicate clearly which results support that affirmation.

**We have clarified:**

"Further, biofilm growth was only visible on columns which were run beyond breakthrough, suggesting that considerable biofouling only started after the loading capacity of the tracer was exhausted" P15, L27

P15, L31-36: This section is slightly speculative; no data provided proofs that the device did not create preferential flowpaths. The manuscript would gain strength if the disturbance caused is acknowledged, but at the same time indicating that compared with other implementations the present method could cause lower disturbance.

Was completed/changed to:" Disturbing the natural structure of the sediment, potentially resulting in artificial flow paths is intrinsic to all intrusive techniques, including HPFM. Still, dispensing of a well screen improves the integration of the HPFM in the natural system and prevents the generation of preferential flow paths along the wall of the well screen. Additionally, the HPFM include a measurement time that is long relative to the duration of the installation, suggesting that the presented method causes lower disturbance compared to other intrusive measurements." P16, L6ff

P15-16, L29-58, L1-9: What about the impact caused the presence of material of different permeability in the hyporheic zone? Were the permeability or porosity of the resin and AC similar to the conditions found in the stream? Structures of different permeability of the surroundings influence hyporheic flows (Ward et al., 2011). Not only the mesh, but the properties of the resin could influence the results, and create preferential flows.

As indicated in section 2.5.2. this challenge was addressed in earlier studies by Klammler et al. (2004) and Hatfield et al. (2004). The correction term for divergence/convergence around the device does account for the permeability of the HPFM matrix in relation to the permeability of the surrounding, also also mentioned in chapter 2.5.2. However, we agree that the effect of heterogeneous permeability of the hyporheic zone has to be mentioned here.

We therefore added: "Heterogeneous permeability of the hyporheic zone around the HPFM does not distort the correction term as long as the permeability of the surrounding media is substantially lower than the permeability of the HPFM matrix. Pre-measurements are therefore necessary for the selection of a suitable resin and tracer carrier. "P16 L15ff

P16, L5: It would be interesting to provide an estimation of the minimum time of exposition for the HPFMs, although it would of course depend on nutrient concentration and fluxes. Would it be possible to estimate it using for example the interstitial velocities measured by Angermann et al. (2013)? Angermann, L.; Krause, S.; Lewandowski, J., Application of heat pulse injections for investigating shallow hyporheic flow in a lowland river. Water Resources Research 2012, 48, W00P02.

The velocities measured by Angermann et al. were in the range of  $1 - 4 \ge 10^{-5}$  m/s, which is around 80 to 350 cm/day. We provide an example for velocities of 20 - 200 cm/day." Based on the accumulated values detected in the control, the minimum deployment time can be estimated. In systems with high

nutrient concentrations, usually the flow velocity  $q_x$  will be the limiting factor. In our application the MDL for  $q_x$  derived from the control was 8.4 cm for the complete deployment time (7 days). If the method inherent uncertainty should not be more than 5 % of the total measurement, the product of duration (in days) and velocity (in cm) should be at least 168 (20 times 8.4). As an example: if measured  $q_x$  is around 200 cm d-1, one day (24 h) of deployment is sufficient. The lowest  $q_x$  detected in our assessment was 21 cm d-1 (in HPFM AC4, see figure 3f) that actually a deployment duration of 8 days would have been optimal. The same estimation can additionally be derived for expected nutrient fluxes. In systems with low nutrient concentrations, it would be recommendable to start estimating the minimum deployment time based on the nutrient fluxes. It should be mentioned here that the suggested calculations assume that the values detected in the control are related to tracer losses during installation and retrieval and analytical uncertainties during sample processing (which is to say that tracer loss during storage is negligible)." P14, L33ff

Also see the added line in the result section, describing the results from the control:" Over the 7 days duration of the experiment, accumulated horizontal flow velocities of  $q_x = 8.4$  cm 7 d-1 and nutrient fluxes of 29.4 mg NO3--N m2 7 d-1 (std = 0.1 mg m2 d-1) and 36.4 mg SRP m2 7 d-1 (std = 0.9 mg m2 d-1) were detected in the control HPFM. Breaking these results down to dial values, yields  $q_x = 1.2$  cm d-1 and nutrient fluxes of 4.2 mg NO3--N m2 d-1 (std = 0.1 mg m2 d-1) and 5.2 mg SRP m2 d-1 (std = 0.9 mg m2 d-1)." P13, L10ff

An additional question might be, at which flow velocities it still makes sense to measure them or at which point diffusion might be more important than advective flow. However, we did not assess this question, and therefor will not discuss it in our article.

P16, L5-9: Would it be possible to provide an estimation of maximum and minimum permeabilities where the HPFMs could be implemented? For example too different permeabilities compared with surrounding sediment will in overestimated or underestimated fluxes (see Ward et al., 2011). Are there limitations of porosity imposed by the resin or AC producers? Additionally, at very high porosities, the time of exposure might be lower than for lower porosities. While time would highly depend on the nutrient concentrations, it would more limiting water fluxes. Such practical approximations would be of great help for potential users of the HPFMs.

Ward, A. S.; Gooseff, M. N.; Johnson, P. A., How can subsurface modifications to hydraulic conductivity be designed as stream restoration structures? Analysis of Vaux's conceptual models to enhance

hyporheic exchange. Water Resources Research 2011, 47, W08512.

The lower limit (very low porosity) will be determined by the flow velocity (see above). Resins are available in a wide range of porosity. The hydraulic conductivity of the activated carbon is around 300 m/d, which is in the range of coarse sand to gravel. Corrections for higher hydraulic conductivity of the surrounding can be done as described in the article (Chapter 2.5.2.) P9L1ff. We assume that at much higher permeabilities (very coarser sediment), the major restrictions will be related to problems in installing the sampler.

However, we did not assess this question in particular.

P16, L19-20: Heterotrophic respiration occurs in all sediments, the rates vary partly as a function of mass transfer of nutrients. Diurnal oscillations due to benthic primary production can of course favour night denitrification, since the limit oxic layer would oscillate.

We changed this sentence to "Microbial consumption of  $O_2$  in the sediments can, depending on nutrient concentration in surface water and transfer of these nutrients to the sediments, result in  $O_2$  depletion in the subsurface. Especially in nutrient rich streams the related diurnal oscillations in  $O_2$  concentration favor night time denitrification in the hyporheic zone" P16 L32

P16, L20: Denitrification is not the only process affecting the NO3 concentrations; also NO3 uptake for biomass production can influence concentrations.

Correct, but biomass production is not primarily happening during night (not bound to low oxygen concentration) and does therefore not contribute to explaining the observed differences between low early morning concentrations and higher noon concentration of NO3  $\rightarrow$  more uptake during the night. Also we here discuss the effect of oxygen concentration. However, you are right we agree that the use of denitrification versus NO3 uptake was not always thorough enough and corrected/completed the article where appropriate

P17, L29-30: Flowpath length can be defined as the distance travelled by the water before leaving the hyporheic zone (see Findaly, 1995), in that case the flowpath length cannot be "derived from the residence time of water and solutes in the hyporheic zone HZ and the horizontal Darcy velocity".

Instead, the flowpath velocity can be assessed. However, flowpath can also be used as the distance travelled by a solute before being uptaken into the microbial community, something similar to the uptake length in nutrient cycling. Please indicate clearly which term is used. Findlay, S., Importance of Surface-Subsurface Exchange in Stream Ecosystems - the Hyporheic Zone. Limnology and Oceanography 1995, 40 (1), 159–164.

We added: "... requires a flow path length. In the presented example, this length refers to the horizontal vector of the distance the water travels in the subsurface before infiltrating the HPFM. The horizontal vector can be derived from the water residence time..." P18 L6

P17, L33: I am not sure if the difference "difference between the theoretically transported NO3- mass MNO3-HZ theor, which is the product of QHZ and CNO3-SW and the measured mass flux MNO3-HZ real" can be defined as denitrification. Denitrification is the reduction of nitrate to N2, however since no N2 was measured the calculations cannot distinguish between denitrification and nitrate uptake to build biomass.

We agree on this. Here (and where else appropriate) we replaced denitrification by nitrate uptake/removal

P18, L18-19: The limitation of nutrient removal by mass transfer is always the case, in very low permeable sediments mass transfer limits nutrient removal independently of the human activities tacking place in the stream or catchment. I propose another view of this section and for the significance of the HPFMS: knowing how mass transfer influences nutrient removal is crucial to manage streams and rivers, especially in the light of the worldwide increase in morphological alterations (Borchardt and Pusch, 2009), eutrophication (Ingendahl et al., 2009) and sediment loading (Hartwig and Borchardt, 2015).

Adopting your suggestion we changed the section to "Quantitative and qualitative knowledge about the influence of mass transfer on hyporheic nutrient removal is crucial to manage streams and rivers, especially in the light of increasing worldwide morphological alterations (Borchardt and Pusch, 2009), eutrophication (Ingendahl et al., 2009) and sediment loading (Hartwig and Borchardt, 2015)" P18 L32ff

P18, L20: The term "horizontal fluxes" is a bit confusing, since many of the results presented in the manuscript are compared with the surface water concentrations. Horizontal fluxes in the hyporheic zone are flowpaths, which are kind of vertically started in the surface or groundwater. Therefore the direction of the fluxes does not seem relevant, although it is true that the manuscript does not represent all the potential fluxes, the HPFMs could be simply oriented differently to gain information of different directions.

We agree that the term is confusing here and deleted the word "horizontal". Overall we have to admit that what we actually measured was the horizontal vector of the nutrient fluxes, which we clarified at other locations in the article eg P 18 L6. However in this sentence the direction of the flux is not of much importance.

Indeed, flux meters measuring vertical fluxes have to be constructed a bit differently, as presented by Layton 2015. Discussing this in detail however would not be in the scope of this article

P18, L28: Based on the definition of denitrification provided in the introduction (P2, L13) this manuscript does not report denitrification rates since the measurements are based on changes in NO3 concentrations without additional measurements of N2.

**See above, changed to "nitrate uptake"**

Figure 5: It might be interesting to include the values of SW during the MLS sampling.

We replaced the min and max concentrations by the concentrations during MLS sampling, as we agree that those are assumedly more informative and presenting both might be too much information.

**Technical corrections**

P2, L23: If the acronym DEA is not used anywhere in the text, it is not necessary.

**"DEA" deleted**

P12, L23: Replace "average concentration observed in the HPFM" with "average concentrations measured with the HPFM" replaced as suggested

Results: the standard deviations are presented in an uncommon way; it might be easier to read when the values are presented as for example (P13, L15)  $4.2 \pm 0.1$  mg NO3- m2 d-1. Changed as suggested

P13, L28: In the tables dissolved oxygen is named as O2, please keep uniform terms throughout the text. Changed DO to O2

P13, L30: Add (MLS) to the heading as in the methods section. Added

P13, L37: Minima and maxima are the plural or minimum and maximum, however concentration is written in singular.

Changed to concentrations

P17, L15: replace degrease with decrease. corrected

P19, Table 1: Use the same format (italics, subscript for Rd) in the table and in the caption. corrected

**Quantifying nutrient fluxes in Hyporheic Zones with a new Passive Flux Meter (HPFM)**

Julia Vanessa Kunz1\*, Michael D. Annable2, Jaehyun Cho2, Wolf von Tümpling1, Kirk Hatfield2, Suresh Rao3, Diatrich Boreherdt1 Michael Pode1

5 Dietrich Borchardt1, Michael Rode1

1Helmholtz Centre for Environmental Research UFZ, Magdeburg, Germany
 2University of Florida, Gainesville, Florida (USA)

[revised manuscript text omitted]
 04.06.-11.06.2015 and08.10. - 11.10.2015: Temp= temperature, SpC=specific conductivity, O2 =dissolved oxygen